# Motivated misremembering of selfish decisions

Ryan W. Carlson 📧 [1✉], Michel André Maréchal 📧 [2], Bastiaan Oud[2], Ernst Fehr 📧 [2] & Molly J. Crockett 📧 [1✉]

People often prioritize their own interests, but also like to see themselves as moral. How do individuals resolve this tension? One way to both pursue personal gain and preserve a moral self-image is to misremember the extent of one's selfishness. Here, we test this possibility. Across five experiments ($N = 3190$), we find that people tend to recall being more generous in the past than they actually were, even when they are incentivized to recall their decisions accurately. Crucially, this motivated misremembering effect occurs chiefly for individuals whose choices violate their own fairness standards, irrespective of how high or low those standards are. Moreover, this effect disappears under conditions where people no longer perceive themselves as responsible for their fairness violations. Together, these findings suggest that when people's actions fall short of their personal standards, they may misremember the extent of their selfishness, thereby potentially warding off threats to their moral self-image.

[1] Department of Psychology, Yale University, New Haven, CT, USA. [2] Department of Economics, University of Zurich, Zurich, Switzerland. ✉email: ryan.carlson@yale.edu; molly.crockett@yale.edu

Humans tend to see themselves as fair and honest[1,2], and often behave accordingly. Yet some are tempted to stray from such ideals when they can get away with it[3–5]. This suggests that, at least in some cases, people prioritize appearing moral to themselves and others over truly aligning their actions with moral principles[5–7]. This seems to be an effective strategy as well. Maintaining a positive moral self-image not only bolsters psychological well-being and physical health[8–10], but also yields downstream social benefits. Individuals who appear committed to moral rules are viewed as more attractive social partners[11,12]— and one dependable strategy for convincing others of one's morality is to first convince oneself of it[13,14].

How can people preserve their moral self-image while simultaneously behaving selfishly? Social scientists have often credited our ability to engage in motivated reasoning[15]—that is, we form self-serving beliefs and attitudes to justify immoral acts to ourselves before or after the events unfold[7,16]. This feat is accomplished in a number of ways. For one, people tend to strike a justifiable balance between self-interest and their moral values—for instance, lying just enough to profit financially, but not so much as to harm their moral self-image[7]. In addition, people psychologically distance themselves from their unethical actions—attributing past misdeeds to situational pressures[17], or having been a "different person" at the time[18,19]. Moreover, people exploit uncertainty—behaving more selfishly when the consequences for others are ambiguous[20], making self-serving mistakes[21], and avoiding information about how their actions may have harmed others[6,22,23]. A common thread in each of these self-serving strategies is that they operate over abstract beliefs and attitudes.

Another possibility that has received less attention is that our desire to believe we are moral may distort memories of our concrete experiences. When people's actions fall short of their personal standards, they might misremember having acted in line with those standards. Misremembering past immoral behavior as moral would pre-empt the need to rationalize one's actions, as actions that fall short of one's personal standards would instead be revised in memory. This possibility coheres with evidence that people are able to "suppress" awareness of unwanted memories at both encoding and retrieval[24]. It is also consistent with recent evidence that memories for dishonest behavior (e.g., cheating) are less subjectively vivid than memories of honest actions[25], and that memories are less accurate when recounting relevant moral rules after cheating[26,27], relevant story details after hypothetical acts of cheating (Kouchaki and Gino[25], but see also Stanley et al.[28]), and selfish relative to altruistic behavior[29]. However, it remains an open question whether people who violate their own moral standards actually misremember their behavior in a self-serving direction. Importantly, impaired vividness or accuracy (e.g., fuzzy recollection of the moment you tipped a barista yesterday) does not necessarily imply memory distortion (e.g., misremembering tipping the barista more than you actually did), and vice-versa[18,30,31]. This leaves open the question of whether memory distortion may serve as another mechanism through which individuals can act selfishly and ultimately still feel moral. When behaving unfairly (e.g., giving a stingy tip), people may misremember behaving more fairly than they actually were, thus preserving the view that they treat others equitably.

Here, we test this possibility by leveraging experiments in which motivated reasoning should have a minimal influence—recalling a recently performed action for which one's standard of fairness is explicitly declared. If one engages in motivated reasoning before or after engaging in unethical behavior, there should be no reason to misremember a justified action. If people instead tend to engage in motivated misremembering, such biases should be evident at recall.

Another key focus of this work concerns whether misremembering is in fact motivated. A rich, long-standing debate in social psychology concerns whether self-serving biases require motivation[32,33]. For instance, many seemingly 'motivated' social comparative biases (e.g., above-average effects) can arise from rational inference processes[33,34]. Although the moral domain is one where motivated effects are well-supported[7,35], it is nonetheless crucial to rule out the possibility that rational inference processes may produce misremembering that only seems morally motivated. For instance, since people tend to be fair across many social situations, it is reasonable for them to infer they behaved fairly in the past. Thus, if memory for recent behavior is weak or incomplete, people may tend to rely on knowledge of what they normally do to inform recall.

We predicted that motivated misremembering would be specific to fairness "violators", as we posited that this process serves chiefly to reduce discomfort when an individual's actions threaten their moral self-image[36,37]. By focusing on personal standards of fairness in four of our experiments, we accounted for an important feature of social decision-making: even when people's overt behavior appears self-serving, people may not subjectively perceive their behavior as selfish[38]. Crucially, here, we measured each individual's subjective threshold for what counts as a fair (versus selfish) choice, and examined how this subjective threshold shapes motivated misremembering of past social choices.

In line with the predictions above, across two lab experiments (Experiments 1 and 2) and three online experiments (Experiments 3, 4a, and 4b), we find evidence that selfish decisions are more generous in hindsight. After deciding how to split money with anonymous partners, people tend to recall being more generous in the past than they actually were. Crucially, this misremembering effect occurs chiefly among people who initially give less than what they personally believe is fair (Experiments 2, 3, 4a, and 4b). Moreover, this effect disappears under conditions where fairness violations no longer pose a self-threat (Experiment 4b)—supporting the view that this misremembering effect is motivated. Together, these findings suggest that when people perceive their own actions as selfish, they are motivated to misremember having acted more equitably, thereby preserving their moral self-image.

## Results

**Misremembering stingy behavior.** In all experiments, participants made a series of five decisions, in which they chose how to allocate money between themselves and a unique anonymous partner. Later, in a surprise memory test, we asked participants to report how much they remembered giving on average across their five decisions, and financially incentivized them to be accurate. In Experiment 1, we predicted that participants would misremember being more generous than they actually were, and that this effect would be driven by relatively stingy participants (since generous participants should have no reason to misremember their generosity). To test this hypothesis, we examined the direction of participants' memory errors (i.e., discrepancies between recalled generosity and actual generosity). If our hypothesis is correct, then memory errors should tend to be self-serving: that is, recalled generosity should on average be greater than actual generosity.

Across all five experiments, memory errors were non-normally distributed ($W = 0.53\text{-}78$, $p < 0.001$; Shapiro–Wilk test), thus we report non-parametric statistics for all key comparisons. All reported $p$-values are two-tailed.

In line with our predictions, in Experiment 1, we found that participants overall showed a systematic bias toward self-serving

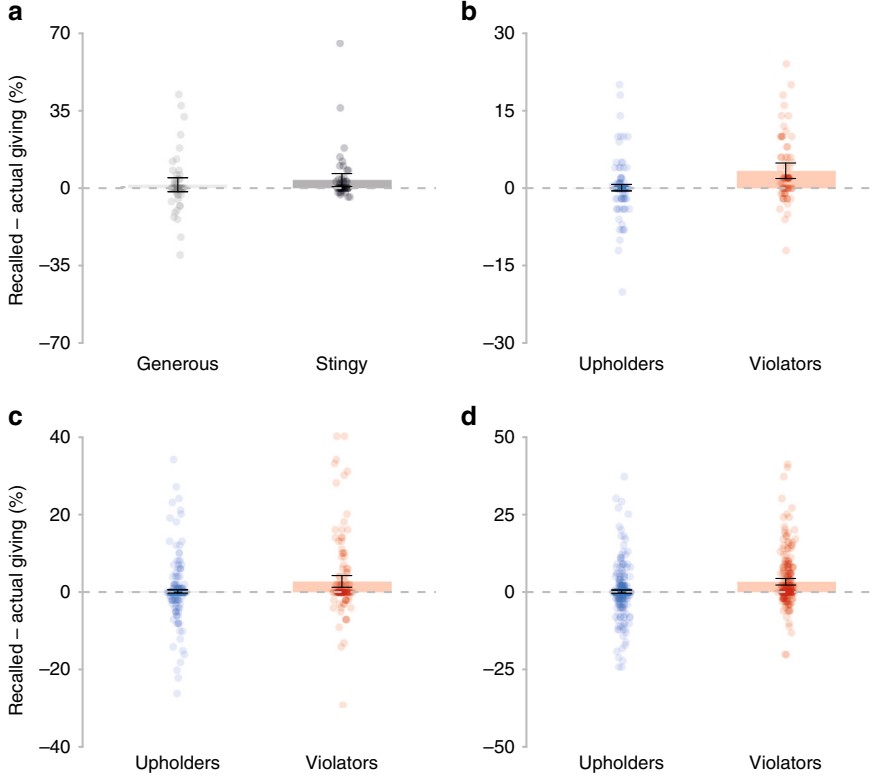

**Fig. 1 Self-serving memory errors.** Differences between the mean percent participants recalled giving, versus how much they actually gave. Positive values reflect "self-serving" memory errors. **a** In Experiment 1, relatively stingy-behaving participants recalled being more generous than they actually were, but not generous-behaving participants. **b** In Experiment 2, we show that specifically individuals who violate their own personal standard of fairness (violators) recall being more generous than they actually were, but not those upholding their own fairness standard (upholders). In Experiment 3 (**c**) and Experiment 4a (**d**), we replicate the finding that violators, but not upholders, recall being more generous than they truly were. Error bars represent 95% confidence intervals (CIs).

memory errors, such that recalled generosity was significantly greater than actual generosity ($V = 1345.5$, $p = 0.021$, $d = 0.23$, Cliff's $\delta = 0.14$; Wilcoxon signed-rank test; means and standard deviations are reported in Supplementary Table 1).

Since we predicted that misremembering would occur chiefly among more stingy participants, we performed a median split based on participants' average level of generosity across their five decisions, dividing them into a behaviorally stingy group ($N = 53$) and a behaviorally generous group ($N = 56$). As predicted, more stingy participants showed a bias toward self-serving memory errors, such that they remembered giving significantly more than they actually did ($V = 335.5$, $p = 0.003$, $d = 0.34$, $\delta = 0.23$; Wilcoxon signed-rank test; Fig. 1a), whereas more generous participants showed no such difference ($V = 353$, $p = 0.54$, $d = 0.13$, $\delta = 0.05$; Wilcoxon signed-rank test). However, stingy participants and generous participants did not significantly differ in the extent of their self-serving memory errors ($W = 1292.5$, $p = 0.23$, $d = 0.19$, $\delta = 0.13$; Mann–Whitney U test).

**Misremembering depends on violating personal standards.** Experiment 1 showed that more stingy individuals recall being more generous than they truly were, while more generous individuals recall their choices accurately. However, the extent of misremembering did not significantly differ between these groups. One possibility is that by only measuring overt behavior, we failed to capture the extent to which people's choices truly upheld, versus violated, their personal standards of fairness for the situation. If misremembering is primarily driven by a desire to reduce threats to one's moral self-image, self-serving memory errors should occur chiefly when people's actions fall short of

their own personal standards, independent of how objectively high or low those standards may be.

In Experiments 2 and 3, we tested this possibility by explicitly probing participants' subjective beliefs about what would count as a fair amount to give, before they made their decisions. We predicted that, independent of how much participants actually gave, misremembering would occur specifically among those participants whose true generosity fell short of what they personally believed was fair.

In addition, we tested whether this effect persists when we introduce a stronger incentive to be accurate. In Experiments 1 and 2, accurate recall was incentivized with a fixed bonus, such that participants received a bonus if their recalled generosity was within 10% of their true average. To up the stakes, in Experiment 3, we offered a monetary bonus that scaled directly with accuracy, such that even a 1% deviation in recalled generosity would reduce the participant's bonus.

Across Experiments 2 and 3, participants again showed a systematic bias towards self-serving memory errors, such that their recalled generosity was significantly greater than their actual generosity ($V_{\text{exp.2}} = 4450$, $p = 0.003$, $d = 0.21$, $\delta = 0.11$; $V_{\text{exp.3}} = 7990.5$, $p = 0.008$, $d = 0.13$, $\delta = 0.05$; Wilcoxon signed-rank tests, see Supplementary Table 1).

To test whether this effect is driven by violating one's personal standards, we separately assessed memory errors in participants who (on average) gave less than what they indicated was fair (violators; $N_{\text{exp.2}} = 69$; $N_{\text{exp.3}} = 143$), versus those who (on average) gave at least as much as what they indicated was fair (upholders, $N_{\text{exp.2}} = 165$; $N_{\text{exp.3}} = 461$). As predicted, violators recalled being significantly more generous than they actually were

($V_{exp.2} = 1396.5$, $p < 0.001$, $d = 0.54$, $\delta = 0.38$; $V_{exp.3} = 1558.5$, $p < 0.001$, $d = 0.30$, $\delta = 0.15$; Wilcoxon signed-rank tests; Fig. 1b, c), but this was not the case for upholders ($V_{exp.2} = 838$, $p = 0.89$, $d = 0.02$, $\delta < 0.001$; $V_{exp.3} = 2439$, $p = 0.69$, $d = 0.04$, $\delta = 0.01$; Wilcoxon signed-rank tests). In addition, when comparing the two groups directly, violators showed a significantly greater bias toward self-serving memory errors ($W_{exp.2} = 3710$, $p < 0.001$, $d = 0.68$, $\delta = 0.35$; $W_{exp.3} = 28390.5$, $p = 0.001$, $d = 0.43$, $\delta = 0.14$; Mann–Whitney $U$ tests).

To confirm that misremembering is driven primarily by violating one's personal standard of fairness, rather than their overt level of giving, we next assessed whether participants' objective level of generosity would still show an influence on memory when excluding violators from analyses. In other words, does misremembering occur among those who give objectively lower amounts, but do not violate their own fairness standard? To test this, we performed a median split on the generosity of only upholders, dividing them into a behaviorally stingy group ($N_{exp.2} = 82$; $N_{exp.3} = 230$) and a behaviorally generous group ($N_{exp.2} = 83$; $N_{exp.3} = 231$; see Supplementary Table 2).

In support of our hypothesis that violating one's personal standards drives misremembering, we found no evidence of a bias toward self-serving memory errors in either behaviorally stingy upholders ($V_{exp.2} = 112$, $p = 0.51$, $d = 0.07$, $\delta = 0.04$; $V_{exp.3} = 1476.5$, $p = 0.11$, $d = 0.14$, $\delta = 0.05$, Wilcoxon signed-rank tests), nor behaviorally generous upholders ($V_{exp.2} = 332.5$, $p = 0.42$, $d = 0.02$, $\delta = -0.04$; $V_{exp.3} = 122.5$, $p = 0.11$, $d = 0.10$, $\delta = -0.02$; Wilcoxon signed-rank tests). Moreover, we found no difference between behaviorally stingy versus generous upholders in their tendency to make self-serving memory errors ($W_{exp.2} = 3179$, $p = 0.39$, $d = 0.09$, $\delta = 0.07$; $W_{exp.3} = 24742.5$, $p = 0.073$, $d = 0.24$, $\delta = 0.07$; Mann–Whitney $U$ tests).

These results provide initial support for the idea that personal standards can influence how people remember their generosity. If this is the case, then recalled generosity should be accounted for not just by actual generosity, but also by the extent to which actual generosity deviates from personal standards. To formally test this possibility, we compared a model predicting recalled generosity from actual generosity, with one predicting recalled generosity from both actual generosity and the deviation between actual generosity and personal standards. Importantly, we controlled for four other predictors of memory in each model: choice speed (the average time participants took to make their choices), choice variance (the standard deviation of their choices), non-giving (whether or not they ever gave a positive amount across their choices), and numeracy (their performance on a numeracy task between making and recalling their choices). As predicted, we found that the model which additionally included fairness deviations accounted for recalled generosity better than the simpler model that did not include fairness deviations as a predictor ($\Delta AIC_{exp.2} = 11.47$, $\Delta BIC = 8.02$, $\chi^2(1) = 13.47$, $p < 0.001$; $\Delta AIC_{exp.3} = 6.33$, $\Delta BIC = 1.92$, $\chi^2(1) = 8.33$, $p = 0.004$; likelihood-ratio tests; Supplementary Tables 5 and 6).

**Misremembering as a motivated phenomenon.** Next, we sought evidence that the misremembering we observe here is in fact motivated (i.e., driven by a desire to preserve one's moral self-image) rather than based on a rational inference (i.e., reconstructed from past behavior based on an inference from their fairness beliefs). If participants, when faced with difficulty recalling their true choices, simply reconstruct those choices based on an inference from what they believe is fair, we would predict that those who gave less than their personal standard (i.e., violators) would tend to recall giving more than they actually did, while those who gave more than their personal standard (i.e., exceeders)

would tend to recall giving less than they actually did. However, if misremembering is in fact motivated, we would only expect misremembering to occur among violators, whereas exceeders should tend to recall their choices accurately. To test these possibilities, across Experiments 2 and 3, we examined memory errors in exceeders ($N_{exp.2} = 69$; $N_{exp.3} = 207$; see Supplementary Table 2)—a group that showed a similar degree of norm deviation and choice variance as violators (see Supplementary Results). In support of our hypothesis that misremembering is motivated, we found no evidence of misremembering toward the norm in exceeders ($V_{exp.2} = 587.5$, $p = 0.48$, $d < 0.001$, $\delta = -0.04$; $V_{exp.3} = 1704.5$, $p = 0.99$, $d = 0.04$, $\delta = 0.01$; Wilcoxon signed-rank tests), and violators misremembered to a significantly greater degree than exceeders ($W_{exp.2} = 1537$, $p < 0.001$, $d = 0.57$, $\delta = 0.35$; $W_{exp.3} = 12767.5$, $p = 0.015$, $d = 0.31$, $\delta = 0.14$; Mann–Whitney $U$ tests).

Nevertheless, even if exceeders show no bias on average, a rational inference account would still find partial support if exceeders were more likely to make errors in the direction of their norm (i.e., norm-directed memory errors) than in the opposite direction. To address this question, we compared the frequency of errors made in each direction by exceeders and violators. For this analysis, we were limited to a smaller sample as we were only able to analyze data from those violators ($N_{exp.2} = 58$; $N_{exp.3} = 64$) and exceeders ($N_{exp.2} = 51$; $N_{exp.3} = 82$) who in fact made memory errors. A binomial test indicated that, whereas the proportion of violators who made norm-directed memory errors was greater than chance ($p_{exp.2} < 0.001$, 95% CI [0.59, 0.83]; $p_{exp.3} = 0.008$, 95% CI [0.54, 0.78]), the proportion of exceeders who made norm-directed memory errors was no different than that expected by chance ($p_{exp.2} = 0.78$, 95% CI [0.33, 0.62]; $p_{exp.3} = 0.91$, 95% CI [0.40, 0.62])). These results are inconsistent with a rational inference account, but consistent with a motivated account of misremembering.

**Motive-dependence and choice-independence.** The findings of Experiments 2 and 3 cohere with the possibility that the memory bias we observe is a motivated phenomenon. However, they only demonstrate this phenomenon indirectly. A motivated account of misremembering would receive compelling evidence over a rational inference account with direct evidence for two key conditions.

First, motivated misremembering should be motive-dependent. That is, misremembering should occur only when people have a motive to misremember, and should cease to occur if motives to misremember are removed. In particular, a rich, long-standing literature on motivated cognition suggests that when people no longer view themselves as personally responsible for a dissonance-inducing action, they no longer experience dissonance motivation[39–42]. Indeed, feelings of responsibility for an action serve as a bridge between the action and one's self-concept[43]. As such, only if people feel personally responsible for their fairness violation should they experience a moral self-threat, and become motivated to reduce it via misremembering.

Second, motivated misremembering should be largely choice-independent. That is, it should be independent of any peculiarities of the choices that violators and upholders tend to make. Indeed, one possible alternative explanation for our findings is that there is something about the pattern of violators' choices that makes their choices more prone to being misremembered—for instance, that the amount they gave across their choices varied more. Thus, it is important to rule out the possibility that the sets of choices made by violators are just inherently more prone to being misremembered than the sets of choices made by upholders.

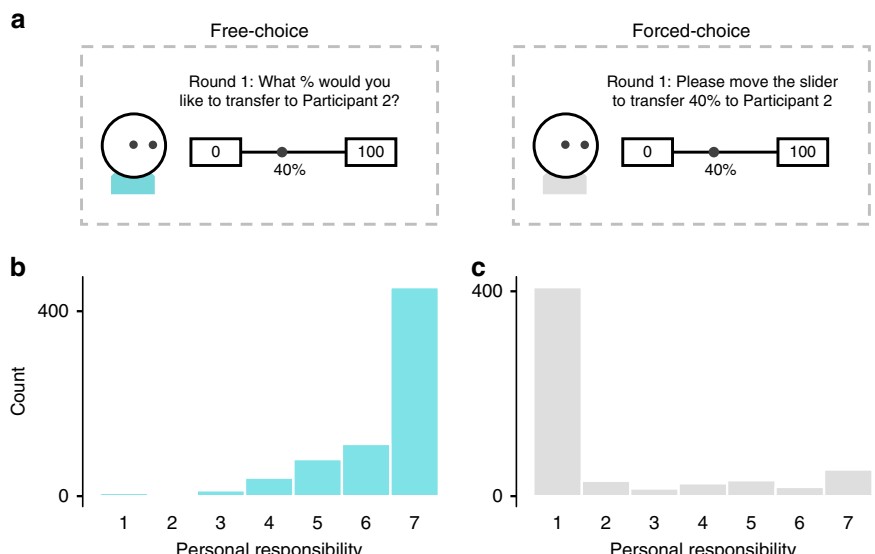

**Fig. 2 Manipulation of responsibility in Experiments 4a and b.** The top panel (**a**) shows an example of a yoked pair of deciders making a free-choice transfer (Experiment 4a) and a forced-choice transfer (Experiment 4b). The amount (and order) of each of the five transfer choices were identical for yoked deciders. The bottom panel shows self-reported personal responsibility from 1 ("Not at all responsible") to 7 ("Extremely responsible") among deciders in Experiments 4a (free-choice; **b**) and Experiment 4b (forced-choice; **c**).

In a final pair of experiments (Experiments 4a and b), we examined these two hypotheses to directly test the motivated nature of misremembering. In Experiment 4a, an initial wave of participants (free-choice deciders) freely decided how to split money with anonymous partners, and later recalled their choices (as in prior experiments). Next, in Experiment 4b, a second wave of participants (forced-choice deciders) were each yoked to a randomly selected free-choice decider from Experiment 4a, and were forced to make (and later recall) the exact same set of transfer choices as their yoked free-choice decider (See Fig. 2a). Crucially, in Experiment 4b, forced-choice deciders (as well as the recipients of their transfers) were instructed that the decider had no responsibility for their transfer choices—thereby removing any incentive to misremember their choices. Moreover, we confirmed that feelings of personal responsibility were lower among forced-choice (versus free-choice) deciders (See Fig. 2b, c for distributions of reported personal responsibility).

We made the following key predictions. Based on the motive-dependence hypothesis, we predicted that those who freely violated their own standards would make self-serving memory errors, whereas those who were forced to violate their own standards and perceived no personal responsibility for their actions would show no such bias toward self-serving memory errors.

Similarly, based on the choice-independence hypothesis, we predicted that while violators in the free-choice condition would make self-serving memory errors, their yoked counterparts in the forced-choice condition, who had to recall exactly the same choices, would not make such errors.

Since Experiment 4a was structured the same as Experiments 2 and 3, we first conducted all key tests that were reported in Experiments 2 and 3. We replicated all of the key results reported above (see Supplementary Results).

**Misremembering is motive-dependent.** To evaluate the efficacy of our manipulation, we first assessed how responsible participants perceived themselves to be for their actions under free-choice (Experiment 4a) and forced-choice (Experiment 4b) conditions. As predicted, free-choice participants reported a high

degree of responsibility for their actions ($M = 6.28$, SD = 1.18), with 64% ($N = 452/709$) reporting the highest level of responsibility possible for their actions (7 = "Extremely responsible"; Fig. 2b). In contrast, forced-choice participants reported a low degree of responsibility for their actions ($M = 2.14$, SD = 2.03), with 70% ($N = 408/580$) reporting the lowest level of responsibility possible for their actions (1 = "Not at all responsible"; Fig. 2c). The difference in ratings between these two groups was significant ($W = 375501.5$, $p < 0.001$, $d = 2.56$, $\delta = 0.83$; Mann–Whitney $U$ test). Nevertheless, some participants in the forced-choice condition indicated they felt some degree of responsibility for their choices despite not making them freely. This is plausible in the context of our experiments because participants ultimately had to still move the slider to register the predetermined choice.

We predicted that while those who freely violated their own standards would make self-serving memory errors, those who were forced to violate their standards (and consequently perceived no personal responsibility for their actions) should show no such bias toward self-serving memory errors.

Consistent with our prior experiments, in Experiment 4a, free-choice participants showed a systematic bias toward self-serving memory errors ($V = 29581$, $p < 0.001$, $d = 0.18$, $\delta = 0.08$; Wilcoxon ranked-sum test). Violators ($N = 231$) recalled being more generous than they actually were ($V = 8609.5$, $p < 0.001$, $d = 0.40$, $\delta = 0.26$; Wilcoxon ranked-sum test; see Fig. 1d), but upholders ($N = 478$) showed no such bias ($V = 6102.5$, $p = 0.92$, $d = 0.02$, $\delta = -0.01$; Wilcoxon ranked-sum test). Furthermore, violators showed a greater bias toward self-serving memory errors than upholders ($W = 42117$, $p < 0.001$, $d = 0.43$, $\delta = 0.24$; Mann–Whitney $U$ test).

By contrast, in Experiment 4b, forced-choice participants did not show a significant bias toward self-serving memory errors ($V = 16087.5$, $p = 0.16$, $d = 0.11$, $\delta = 0.05$, Wilcoxon ranked-sum test; Supplementary Table 1). Further analyses revealed that similar to free-choice upholders, forced-choice upholders ($N = 362$) did not exhibit self-serving memory errors ($V = 3393$, $p = 0.14$, $d = 0.02$, $\delta = -0.02$; Wilcoxon ranked-sum test). Forced-choice violators ($N = 217$)—despite not freely choosing to violate their

their norm—nevertheless did recall on average being more generous than they actually were ($V = 4580.5$, $p < 0.001$; $d = 0.28$, $\delta = 0.15$, Wilcoxon ranked-sum test, Fig 3a). However, the magnitude of misremembering was smaller in forced-choice violators ($d = 0.28$, $\delta = 0.15$) than free-choice violators ($d = 0.40$, $\delta = 0.26$). We also found that forced-choice violators showed a greater bias toward self-serving memory errors than upholders ($W = 33105.5$, $p < 0.001$, $d = 0.34$, $\delta = 0.16$, Mann–Whitney $U$ test), though again this effect was smaller than the difference between free-choice violators and upholders in Experiment 4a ($d = 0.43$, $\delta = 0.24$).

Thus, while our forced-choice manipulation eliminated the main effect of misremembering, there remained a significant, albeit diminished, misremembering effect for forced-choice violators. One possibility is that, despite our instructions that sought to minimize feelings of responsibility in the forced-choice experiment, some forced-choice violators may have still felt some degree of personal responsibility, and thus were motivated to misremember. This prediction follows from work showing that while following orders or instructions can reduce feelings of responsibility, it does not eliminate them entirely (e.g., refs. [40,44]). Supporting our prediction, ~30% of forced-choice participants reported feeling some degree of responsibility for their choices.

Crucially, the motive-dependence hypothesis predicts that the forced-choice manipulation should eliminate motivated misremembering only in those violators who viewed themselves as not personally responsible for their actions, whereas forced-choice violators who viewed themselves as personally responsible should remain motivated to make self-serving memory errors. To test this possibility, we independently assessed the memories of those who reported being "not at all responsible" for their actions (i.e., 1 out of 7 on our personal responsibility measure; $N = 408$), and those who self-reported some degree of personal responsibility for their actions (i.e., greater than 1 out of 7 on our personal responsibility measure; $M = 4.87$, SD $= 1.84$; $N = 171$).

Consistent with the motive-dependence hypothesis, forced-choice deciders who nonetheless perceived themselves as responsible for their actions showed a significant bias toward self-serving memory errors ($V = 3206.5$, $p = 0.019$, $d = 0.20$, $\delta = 0.14$; Wilcoxon ranked-sum test; Supplementary Table 1). More specifically, violators ($N = 79$) showed a significant bias toward self-serving memory errors ($V = 1258.5$, $p < 0.001$, $d = 0.50$, $\delta = 0.46$; Wilcoxon ranked-sum test; Fig. 3c), whereas upholders ($N = 92$) showed no such bias ($V = 372$, $p = 0.066$, $d = 0.08$, $\delta = -0.13$; Wilcoxon ranked-sum test). We also found that violators showed a greater bias toward self-serving memory

errors than upholders ($W = 2010$, $p < 0.001$, $d = 0.61$, $\delta = 0.45$; Mann–Whitney $U$ test).

In contrast, forced-choice participants who reported feeling no responsibility for their actions showed no bias toward self-serving memory errors ($V = 4802.5$, $p = 0.68$, $d = 0.03$, $\delta = 0.01$; Wilcoxon ranked-sum test; Supplementary Table 1). In particular, neither violators in this group of participants ($N = 138$; $V = 874$, $p = 0.61$, $d = 0.05$, $\delta = -0.02$; Wilcoxon ranked-sum test; Fig. 3b) nor upholders ($N = 270$; $V = 1600.5$, $p = 0.93$, $d = 0.03$, $\delta = 0.02$, Wilcoxon ranked-sum test) showed any bias toward self-serving memory errors. Moreover, violators and upholders did not differ in the extent of their self-serving memory errors ($W = 19171.5$, $p = 0.57$, $d = 0.01$, $\delta = 0.03$; Mann–Whitney $U$ test).

Crucially, forced-choice violators who felt responsible reported significantly larger self-serving memory errors than forced-choice violators who did not feel responsible ($W = 3222$, $p < 0.001$, $d = 0.60$, $\delta = 0.41$; Mann–Whitney $U$ test). Moreover, free-choice violators in Experiment 4a made significantly larger self-serving memory errors than forced-choice violators in Experiment 4b who did not feel responsible ($W = 19925.5$, $p < 0.001$, $d = 0.44$, $\delta = 0.25$; Mann–Whitney $U$ test).

**Misremembering is choice-independent**. We also predicted that while those who freely violate their own standards should tend to make self-serving memory errors, those forced to make the same choices would show no bias toward self-serving memory errors—regardless of whether those choices violated their own standards or not. To test this, we directly compared the recall performance of 579 matched pairs of free-choice deciders and forced-choice deciders.

When we examined the subset of Experiment 4a free-choice deciders who were yoked with Experiment 4b forced-choice deciders ($N = 579$), we again observed a systematic bias toward self-serving memory errors in free-choice deciders ($V = 19876$, $p < 0.001$, $d = 0.19$, $\delta = 0.09$; Wilcoxon ranked-sum test; Supplementary Table 1). In particular, violators ($N = 181$) recalled being more generous than they actually were ($V = 5627.5$, $p < 0.001$, $d = 0.42$, $\delta = 0.26$; Wilcoxon ranked-sum test; Fig. 4a), whereas upholders ($N = 398$) showed no such bias ($V = 4255.5$, $p = 0.53$, $d = 0.04$, $\delta = 0.01$; Wilcoxon ranked-sum test). We also found that violators showed a greater bias toward self-serving memory errors than upholders ($W = 27721.5$, $p < 0.001$, $d = 0.44$, $\delta = 0.23$; Mann–Whitney $U$ test).

In contrast, yoked forced-choice deciders—who were forced to make, and then remember, the exact same choice sets as free-

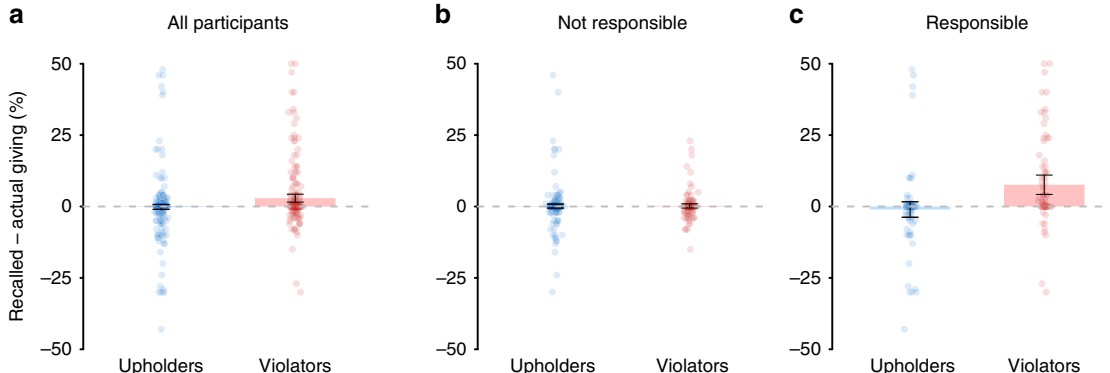

**Fig. 3 Motive-dependence of self-serving memory errors. a** In Experiment 4b, forced-choice violators exhibited more self-serving memory errors than forced-choice upholders. But crucially, this effect depends on whether violators felt responsible for their fairness violations. **b** Forced-choice violators who perceived no responsibility for their actions showed no self-serving memory errors. **c** In contrast, forced-choice violators who nonetheless still perceived themselves as responsible for their actions exhibited self-serving memory errors. Error bars represent 95% CIs.

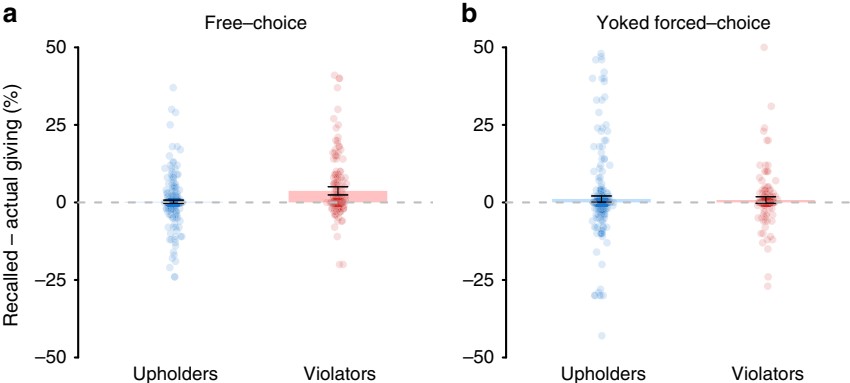

**Fig. 4 Choice-independence of self-serving memory errors. a** In Experiment 4a, free-choice violators exhibited self-serving memory errors. **b** In contrast, in Experiment 4b, participants forced to make identical choices as violators exhibited no such bias toward self-serving memory errors. Error bars represent 95% CIs.

choice deciders—showed no significant bias toward self-serving memory errors ($V = 16087.5$, $p = 0.16$, $d = 0.11$, $\delta = 0.05$; Wilcoxon ranked-sum test; Supplementary Table 1). Indeed, neither those forced to make and then recall the same choices as violators ($N = 181$; $V = 1773.5$, $p = 0.17$, $d = 0.11$, $\delta = 0.12$; Wilcoxon ranked-sum test) nor upholders ($N = 398$, $V = 7149.5$, $p = 0.53$, $d = 0.11$, $\delta = 0.02$; Wilcoxon ranked-sum test) showed self-serving memory errors (Fig. 4b). While those paired with violators tended to have less accurate memories in general ($W = 23396.5$, $p < 0.001$, $d = 0.43$, $\delta = 0.35$; Mann–Whitney $U$ test), memory inaccuracy in forced-choice violators did not predict self-serving memories in yoked free-choice violators (see Supplementary Results). Moreover, crucially, we found no difference in the extent to which yoked forced-choice deciders paired with violators (versus upholders) made self-serving memory errors ($W = 33476$, $p = 0.13$, $d = 0.04$, $\delta = 0.07$; Mann–Whitney $U$ test).

## Discussion

In *The Descent of Man*, Darwin wrote "A moral being is one who is capable of reflecting on his past actions and their motives—of approving of some and disapproving of others"[45]. Here we offer one account of how people may fall short of Darwin's moral criterion. Across five experiments, we found that people tend to recall being more generous than they actually were. This occurred even when participants could receive a monetary bonus that directly scaled with their memory accuracy. Crucially, this effect was driven by a subset of individuals whose generosity fell short of what they personally believed was fair, independent of their absolute level of giving. Such individuals not only showed more self-serving memory errors, but were also less confident in their memories compared to individuals who upheld their personal fairness standards. Further analyses revealed that this phenomenon is in fact motivated, as opposed to arising from rational inference processes. Specifically, misremembering only occurred when violating fairness standards posed a self-threat—i.e., when people perceived themselves as personally responsible for their actions. Together, our findings highlight the importance of assessing selfishness as it is perceived by social decision-makers— that is, relative to their own, subjective moral beliefs. More crucially, these findings contribute to a growing literature on the motivated nature of memory[25–29,46–48] by supporting the idea that people can misremember not just rules, or hypothetical situations, but also concrete actions.

Specifically, these findings suggest that those who violate (as opposed to uphold) their personal standards misremember the extent of their selfishness. Moreover, they highlight the key motivational role of perceived responsibility for norm violations —consistent with classic accounts from social psychology[42], and recent evidence from experimental economics[29]. However, since we focused specifically on those who reported no responsibility, it is also conceivable that other factors might have differed between the participants who felt responsible and those who did not.

We interpret these results as evidence of motivated memory distortion, however, an alternative account would hold that these individuals were aware of their true level of generosity at recall, yet were willing to pay a cost to claim having been more generous. While this account is not inconsistent with prior work[7], it should be less likely in a context which is anonymous, involves no future interaction with any partners, and requires memories to be verified by an experimenter[49]. Accordingly, we found little to no effect of trait social desirability on peoples' reported memories. Together, these points suggest that people were actually misremembering their choices, rather than consciously lying about them.

A related literature suggests that memory is enhanced for emotional events[50]—and violating fairness standards may immediately induce negative emotions such as guilt[51,52]. From this literature, one would expect fairness violations to be more memorable than morally-consistent choices[28]. This may be especially true of extreme moral transgressions (e.g., physical harm), which may remain memorable to transgressors long after they occurred[53], or when one's moral transgression was witnessed by others—which could elicit social emotions such as shame or embarrassment[54]. Such contexts may indeed elicit a different self-serving strategy. If one's moral conduct were witnessed by, or directly comparable to, the conduct of one's peers, individuals might instead distort their moral view of others (e.g., viewing peers as being less moral than they truly are[46,55]). Crucially, here we focused on relatively minor fairness violations, which occurred anonymously. As such, we anticipated that the intensity of negative emotion experienced by agents would not be strong enough to elicit preferential encoding of their own actions—a prediction supported by the presence of self-serving memory errors across our five experiments. Ultimately, uncovering how memory for selfishness shifts as a function of (i) the degree of harm caused by the choice and (ii) the presence of others remain exciting and important directions for future work to explore.

If we accept the possibility that people can misremember their misdeeds, what cognitive and neural mechanisms might facilitate this effect? Motivated misremembering could plausibly emerge at any stage of memory (encoding, consolidation, or retrieval)[24].

Below we propose three possibilities, however, it is important to note that it is plausible that numerous memory mechanisms could contribute to misremembering in our experiments[24,56,57].

One possibility is that selfish individuals misremember their choices via mechanisms that unfold at encoding. Memory encoding is chiefly determined by attention[58], which in turn is guided by our goals[59]. Upon first becoming aware of the intention to make a selfish choice, individuals may attend less to episodic details, and strategically downregulate the depth of processing of such episodes via inhibitory control processes[24]. As a consequence, violators may misremember at recall due to their most selfish choices being less available in memory[60,61]—that is, selfish choices are less richly encoded, and thus more generous choices loom larger at recall.

Another possibility is that selfish choices are accurately stored in memory, but later distorted via retrieval-related processes[24,35]. To this end, individuals might suppress retrieving their more selfish choices. Engaging in this form of motivated control over memory—suppression-induced forgetting—has been shown to not only prevent the suppressed memory from entering conscious awareness, but also reduce its accessibility during subsequent retrieval attempts[62,63]. Such findings suggest that even if selfish and fair choices are stored with equal fidelity, selfish choices may nonetheless become less accessible over time, leading one to preferentially recall choices that better align with one's moral self-view. This raises at least one key direction for future work: since misremembering was more likely to occur when people varied in their choices, it could be strategic to vary the specific amount one gives each time to facilitate motivated misremembering.

Importantly, these cognitive mechanisms of misremembering are neurally dissociable[24,64,65]. For instance, an encoding account of misremembering would be more consistent with decreased functional connectivity between midbrain dopaminergic targets and hippocampus immediately following selfish choices[64]. In contrast, a retrieval account of misremembering would predict increased dorsolateral prefrontal activation and reduced hippocampal activation beyond the encoding phase, reflecting retrieval-related top-down control over memory[62,66]. Teasing apart the influences of such mechanisms may be fruitfully pursued by examining how neural representations of ethical and unethical memories change over time[67]. Such work could allow for a deeper understanding of the mechanisms by which memory operates in the service of one's goals, and specifically, how the malleability of memory helps us maintain ethically spotless minds.

## Methods

**Experiments 1 and 2: lab participants**. In all, 112 participants (57 female, 50 male, 5 did not specify; mean age = 22.0), completed Experiment 1, and 243 participants (118 female, 125 male; mean age = 22.8) completed Experiment 2, respectively. In both experiments, participants were recruited from the University of Zürich participant database. We focused our recruitment on individuals who had not previously participated in experiments involving dictator games. Though, due to an administrative oversight in Experiment 2, there were five participants who did not meet this criterion but nonetheless participated in the experiment, which was discovered after data collection. These five participants were excluded from further analysis.

In Experiment 1, three participants were excluded for being extreme outliers (i.e., >4 standard deviations from the mean) in the size of their memory errors—our key dependent measure. This left a total sample of 109 participants. In Experiment 2, four participants were excluded for being extreme outliers (i.e., >4 standard deviations from the mean) in the size of their memory errors, leaving a total sample of 234 participants. To test the robustness of our results, we confirmed that our findings remained consistent when we additionally exclude subjects who never chose to give money in our task ($N_{exp.1} = 18$; $N_{exp.2} = 51$), as well as when we include all participants in our analyses (see Supplementary Results). The University of Zurich Ethics Commission approved the procedures in Experiments 1 and 2. Both experiments complied with all relevant ethical regulations for work with human participants and all participants provided written informed consent.

**Experiments 1 and 2: lab procedures**. Experiments 1 and 2 were conducted in the Experimental Laboratory at the Department of Economics, University of Zurich. In Experiment 1, data were collected across five experimental sessions, with the number of participants per session ranging from 11 to 30. In Experiment 2, data were collected across 13 experimental sessions, with the number of participants per session ranging from 10 to 31. Participants provided informed consent and then received instructions for the experiment. They were instructed that their responses would remain confidential and anonymous and that they would receive a payment of CHF 10 for participation in the experiment, as well as an additional payment based on one of their choices during the experiment, which they would receive in cash at the end of the session.

In both experiments, participants were instructed that they would be playing a dictator game in which they would be paired anonymously with another participant. They were further instructed that one of the participants (Participant 1) would decide how to divide a sum of money between themselves and the other participant (Participant 2), and were informed that their role would be revealed to them at a later point in time.

One key difference between Experiment 1 and Experiment 2 is that, at this point in Experiment 2, we probed participants' fairness standards by asking them to indicate what they believed to be the "maximum acceptable share" for Participant 1 (the decider) to keep. Half of participants indicated their fairness standards before learning they were assigned the role of Participant 1, and the other half indicated their fairness standard after learning their role. The timing of the fairness standards question had no influence on the results, and thus we collapsed across these conditions in all analyses for Experiment 2 (see Supplementary Results for more details).

In both experiments, participants were ultimately assigned to the role of dictator (Participant 1) for a series of five modified dictator games, each with a unique anonymous partner. As dictators, participants made choices about an endowment that could range in size from CHF 10 to CHF 30, however its size was unknown at the time of choice. For each of their five choices, participants had to select what percentage of the endowment they would like to keep for themselves, and what percentage they would transfer to the anonymous recipient, in 10% increments ranging from [Keep 100%, Send 0%] to [Keep 0%, Send 100%]. They were informed that one of their choices would be randomly selected and implemented, and that the size of the endowment would also be randomly determined and revealed at the end of the experiment. Importantly, it was emphasized to participants that there would be no further interaction with the recipient after making their choices.

Choices were made consecutively on separate screens (for further details on choice format, see Supplementary Methods). After the participant selected and confirmed their choice, the screen advanced to the next choice after a 1-s delay. After making their five choices, participants completed demographic measures (age, gender, and education) as well as a test of numeracy[68]. The numeracy test took ~5 min, ensuring that participants' previous dictator game choices would no longer be in their short-term memory[69].

Finally, participants were presented with a surprise incentivized memory test in which they were asked to recall their previous choices and indicate what percent of the endowment, on average, they transferred to the recipient. Participants were asked to record their response on a provided answer sheet. To motivate participants to recall their choices accurately, they were informed that they would receive an additional CHF 5 if their response was within 10% of their actual average transfer. This setup ensured that participants had no incentive to consciously lie. At the recall phase, participants knew that there would be no future interactions with any partners. They also knew that their reported memories were accessible by the research team—who also had access to the participants true choices in order to determine their accuracy bonus. Such knowledge is known to reduce lying[49].

**Experiments 3: online participants**. In Experiment 3, we determined our sample size by focusing on achieving sufficient power within our main groups of interest: violators and upholders. For Experiment 3, a power analysis showed that a minimum sample size of $N = 128$ would be needed to attain ~95% power to detect a small to medium-sized effect ($d \approx 0.30$) at an alpha level of 0.05. We predicted that at least 25% of participants would violate their personal standard of fairness based on Experiment 2, and thus determined that a minimum total sample size of $128 \times 4 = 512$ participants would be needed.

In all, 647 participants (344 female, 301 male, 2 did not specify; mean age = 36.4) completed Experiment 3. All participants were recruited from Amazon Mechanical Turk. Since this experiment was conducted online, we also included a series of comprehension checks and attention probes (including transcribing several sentences of text, as well as two binary questions in which we asked participants if they understood key aspects of the task). In total, 28 participants were excluded for failing at least one of our comprehension checks. Moreover, five participants were excluded for reporting suspicion about key aspects of the task. Finally, 11 participants were excluded for being extreme outliers (i.e., >4 standard deviations from the mean) in the size of their memory errors. This left a total sample of 604 participants. Our findings remain consistent when we also exclude subjects who never chose to give money in our task ($N = 100$), as well as when we include all participants in our analyses (see Supplementary Results).

The Yale University Human Investigation Committee approved the procedures for Experiment 3. The experiment complied with all relevant ethical regulations for work with human participants and all participants provided informed consent.

**Experiments 3: online procedure**. The experimental protocol for Experiment 3 was largely identical to Experiment 2, with a few exceptions. Like Experiments 1 and 2, choices were made consecutively on separate screens. Before each choice screen, participants saw an intermediary screen for 3 s that said "Round # will now begin."

As dictators, participants made choices about an endowment that could range in size from 10 to 30 cents. Like before, the size of this endowment was unknown at the time of choice.

One crucial difference from prior experiments was how we incentivized the surprise memory test: participants were informed at the test that they would receive a scaling monetary bonus for accurately recalling their generosity, such that they would lose 2% of their bonus for each 1% their recalled average transfer rate deviated from their actual average transfer rate. Unlike the prior experiments, participants were not informed of the size of this bonus until after the experiment. In addition, all participants in this experiment made fairness judgements without prior knowledge of whether they would be assigned as Participant 1 (the decider) or Participant 2 (the receiver). Here, we probed fairness standards by asking participants to report what they believed to be the minimum acceptable share for Participant 1 (the decider) to transfer to Participant 2 (the receiver).

In Experiment 3, participants reported the percentage they recalled transferring on a sliding scale (the "Slider" Question type and "Bars" subtype in Qualtrics). They could report their recalled average in 1% increments on this scale. Furthermore, we used a slightly different scale/depiction from the format in which choices were made. Choices were made also using a similar but visually distinct slider scale (the "Slider" question type and "Sliders" subtype in Qualtrics), and were instead made in 10% increments.

**Experiments 4a and b: online participants**. In Experiment 4a and b, we determined our sample size by focusing on achieving sufficient power within our main group of interest: violators. A minimum sample size of $N = 298$ ($N = 149$ per group) was needed to attain ~95% power to detect the key group difference observed in Experiment 3 ($d \approx 0.43$) at an alpha level of 0.05. We predicted that at least 25% of participants would violate their personal standard of fairness based on our prior experiments, and thus determined a minimum total sample size of $149 \times 4 = 596$ participants was needed.

In total, 1152 participants (469 female, 678 male, five did not specify; mean age = 34.2) completed Experiment 4a, and 1036 participants (571 female, 462 male, three did not specify; mean age = 35.9) completed Experiment 4b—including an initial 719 participants, and an additional 317 participants whose data were collected (within 3 days of the initial data collection period) to ensure sufficient power for paired analyses after exclusions. All participants were recruited from Amazon Mechanical Turk.

In Experiment 4a, we excluded 266 participants who failed at least one of our comprehension checks, as well as 163 participants who reported any suspicion about key aspects of the task. Finally, 14 participants were excluded for being extreme outliers (i.e., >4 standard deviations from the mean) in the size of their memory errors. This left a total sample of 709 participants in Experiment 4a. Importantly, our key findings remain the same when also excluding non-givers ($N = 78$), as well as when including all participants in the analyses (see Supplementary Results). The choices made by each of these 709 free-choice participants formed the 'choice sets' that were subsequently provided to forced-choice participants in Experiment 4b.

In Experiment 4b, we excluded 268 participants who failed at least one of our comprehension checks, and 181 participants who reported suspicion about key aspects of the task. Finally, eight participants were also excluded for being extreme outliers (i.e., >4 standard deviations from the mean) in the size of their memory errors. This left a total sample of 579 participants in Experiment 4b, and ultimately 579 unique yoked pairs of free-choice and forced-choice deciders—though our key findings again remain the same when we run our analyses using all 709 unique yoked pairs, and when excluding yoked pairs of non-givers (see Supplementary Results). The Yale University Human Investigation Committee approved the procedures for Experiments 4a and 4b. Both experiments complied with all relevant ethical regulations for work with human participants and all participants provided informed consent.

**Experiments 4a and b: online procedures**. The procedure of Experiment 4a was nearly identical to Experiment 3. Participants were informed of the allocation task, reported their fairness standard, and then made five choices about how to allocate an endowment of money between themselves and anonymous partners. Later, they received a surprise memory test and were paid for accuracy using the same incentive scheme as in Experiment 3.

In Experiment 4b, participants initially received the same instructions as those Experiment 4a—including the same details for the allocation task. This was to ensure that they reported their fairness standard under the assumption of free-choice (as in Experiment 4a). After reporting their fairness standard, they were

subsequently informed that they had been assigned to complete an alternate version of the task in the role of Participant 1. Specifically, participants were instructed that in the alternate version of the task, they would not be able to choose what proportion of the endowment to transfer to Participant 2, and that transfer amounts would instead be randomly predetermined. On each transfer trial, participants used the same slider to make their transfer choices, however they were instructed to "Please move the slider to transfer X% to Participant 2 this round". To advance to the next round, participants had to move the slider to the specified amount. Crucially, the amount displayed on each of the five trials matched the exact amount and order of transfers made by a yoked free-choice participant from Experiment 4a. After making their allocation choices, the experiment proceeded identically to Experiment 4a. After completing the surprise memory quiz, participants in Experiments 4a and b completed a measure of memory confidence (as in Experiment 3), a measure of personal responsibility, and exploratory measures of psychological discomfort.

**Memory measures: all experiments**. Of central interest in all experiments was the difference between the actual average percentage each participant gave, and their memory of the average percentage they gave. To address this, we conducted two complementary analyses:

We were specifically interested in whether people would remember giving more than they actually gave—that is, if they would make self-serving memory errors. We operationalize motivated misremembering as the occurrence of self-serving memory errors in our task. These two terms are used interchangeably throughout the paper. We measured this tendency by taking into account the direction of participants' memory errors: positive values indicated a participant recalled giving more than they actually gave (i.e., a self-serving memory error), and negative values indicated that they recalled giving less than they actually gave (i.e., a self-defeating memory error).

In addition, we examined memory inaccuracy. To do so we computed the absolute difference between participant's actual and recalled generosity (i.e., the absolute size of memory errors). This allowed us to assess which group tended to make larger memory errors (independent of whether they remembered giving more or less than they actually gave). Results for this measure are reported in the Supplementary Information.

**Control measures: all experiments**. We also assessed several factors that we predicted would impact memory inaccuracy: (i) choice speed—the average reaction time of each participant's five choices, (ii) choice variance—the standard deviation of their five choices, (iii) non-giving—whether or not they ever gave a positive amount across their five choices, and (iv) their performance on the numeracy test described above. Our key results remain the same controlling for these factors, as such we report analyses of these measures in the Supplementary Information (see Supplementary Tables 3 and 4).

Since choice variance could directly impact the memorability of one's choices, we also confirmed that all key findings remain consistent when excluding 'static' deciders—that is, those participants who always made identical allocation choices (See Supplementary Results and Discussion).

**Personal responsibility: Experiment 4a and b**. To assess personal responsibility in Experiments 4a and b, we asked people how responsible they felt for the amount of money that was transferred to the receiver on a scale ranging from 1 ("Not at all responsible") to 7 ("Extremely responsible").

**Internal validity of fairness standards measure: Experiments 2, 3, and 4a**. To support the validity of our measure of personal standards of fairness, we tested how well this measure tracked with people's actual behavior. People's reported standard of fairness were highly correlated with their actual behavior in Experiment 2 ($r_s = 0.78$, $p < 0.001$), Experiment 3 ($r_s = 0.55$, $p < 0.001$), and Experiment 4a ($r_s = 0.72$, $p < 0.001$)—suggesting that such standards reflected a meaningful guide for people's actual behavior (see Supplementary Text for additional analyses). Since participants did not freely make choices in Experiment 4b, we did not test internal validity in this experiment.

**Socially desirable responding: Experiment 3**. In Experiment 3, we probed participants' tendency to engage in socially desirable responding using a short-form (10-item) version of the Marlowe-Crowne Social Desirability Scale[70]. This measure allowed us to rule out the possibility that motivated misremembering can be reduced to socially desirable responding. Our results remain the same when controlling for this factor (See Supplementary Table 6).

**Memory confidence: Experiments 3, 4a, and 4b**. After participants recalled their average generosity, we asked them to rate how confident they were in their recalled generosity on a 7-point Likert scale ranging from 1 ("Not at all confident") to 7 ("Extremely confident"). Spearman rank correlation tests showed that self-reported memory confidence was negatively correlated with the size of participants' actual memory errors in Experiment 3 ($r_s = -0.64$, $p < 0.001$), Experiment 4a ($r_s = -0.55$, $p < 0.001$), and Experiment 4b ($r_s = -0.57$, $p < 0.001$).

**Psychological consequences: Experiments 3, 4a, and 4b.** In Experiment 3, we explored two factors that could shift as a function of whether an individual engaged in motivated misremembering or not: their affective state and their moral self-view. Moreover, in Experiments 4a and b, we explored an additional factor that could shift as a function of whether an individual engaged in motivated misremembering or not: psychological discomfort. For further details, see Supplementary Methods and Results.

**Reporting summary.** Further information on research design is available in the Nature Research Reporting Summary linked to this article.

## Data availability

Raw data for all experiments are publicly available in an Open Science Framework (OSF) repository (osf.io/pzwt7/; [DOI 10.17605/OSF.IO/PZWT7]). A reporting summary for this Article is available as a Supplementary Information file.

## Code availability

Analysis code for all experiments are also publicly available in the same OSF repository (osf.io/pzwt7/).

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

## Acknowledgements
We thank Marcia Johnson for providing thoughtful comments on earlier versions of the manuscript. We also thank members of the Crockett Lab—and especially Jen Siegel, Vlad Chituc, Daniel Yudkin, Clara Colombatto, and Xanni Brown—for scientific discussions and helpful feedback on this work.

## Author contributions
R.W.C., M.A.M., E.F., and M.J.C. designed research; R.W.C., M.A.M., B.O., and M.J.C. conducted the experiments; R.W.C. analyzed data and R.W.C., M.A.M., E.F., and M.J.C. discussed and interpreted the data. R.W.C. wrote the paper under the supervision of M.J.C.—with critical input from M.A.M. and E.F.

## Competing interests
The authors declare no competing interests.
