## [Peer Review File · Nature Communications]

Reviewers' comments:

Reviewer #1 (Remarks to the Author):

Review of Carlson et al., "Motivated misremembering"

Review by R. Baumeister

I quite like this paper. It reports a nice series of experiments with consistent results that make an important point. It makes a valuable contribution to both the morality and the self-deception literatures. I have relatively little to criticize. The paper does use median splits, which purists mostly do not respect any more, but given that they were done for supplementary exploring I think they are fine.

One methodological question I had was how the participants experienced playing the games and making the allocations. Were these simply presented all at once? More to the point, why would someone make a different allocation in different games? It would seem that the rational strategy would be to decide what is the optimal allocation and just give the same answer for all five games. Some data on whether participants did that or not would be helpful. Obviously this is quite relevant to the memory data. If I decide "I will give the other person 30%" and do so for all five games, it would be surprising if I had trouble remembering this.

That brings up another thought: Did people perhaps help themselves forget by giving a variety of different responses, only some of which were selfish and unfair?

On page 23, the authors note an alternative explanation, which is that people who were unfair secretly did know it but did not want to admit it and were willing to sacrifice the accurate memory bonus so as not to admit it. They reject this based on three findings, but I do not find this entirely convincing. First, they say these people had lower memory accuracy. But isn't that the same finding, that they distorted their memory for the sake of making themselves seem more generous? Second, they say these people had lower confidence in their memory. But that would be true for lying, I would think. If I lie to say I gave 40% when I really only gave 20%, and then I'm asked how sure I am about the 40%, I might protect myself by expressing low confidence in case I'm found out. Last, they did not score differently on the the Marlowe-Crowne social desirability scale, but this is not strong evidence. Anyway, I am inclined to agree with the authors' interpretation that actual misremembering is more likely than overt lying (though there is probably a gray area), but the evidence does not seem convincing.

In any case, I think this paper deserves publication in a highly prestigious outlet. My compliments to the authors.

Reviewer #2 (Remarks to the Author):

This is an interesting paper. It joins a long history of research in psychology trying to understand how people maintain consistency between attitudes and behavior, and how they might react to inconsistency. For the better part of 3 decades, social psychologists in particular tested competing accounts for how people maintain consistency, especially testing between motivated reasoning

accounts (such as cognitive dissonance theory; Festinger & Carlsmith, 1957) and rational inference accounts (such as self-perception theory; Bem, 1972). Great care was taken during those decades to both carefully propose a model by which either goal-directed reasoning or simple cognitive inference could guide judgment, and tests of intervening mechanisms were carefully conducted. One of the things we learned during these decades is that it's easier to imagine that people are engaging in motivated reasoning than to actually demonstrate they are doing so. Many beliefs that seem self-serving are not produced by motivated reasoning, as is so easy to assume for anyone with either no familiarity with our field's history or with just a passing familiarity with it. The tendency to rate oneself as above average on many desirable traits, for instance, is easily presumed to be a self-serving judgment driven by motivated reasoning, but the evidence instead suggests that purely cognitive inference mechanisms largely produce these effects (Kruger, 1999; Chambers & Windschitl, 2004). We have come a long way in understanding the mechanisms that guide human judgment.

Unfortunately, it does not seem that the authors of this manuscript are especially aware of this large literature, or are very careful in either articulating or testing their motivated reasoning account. The authors present several experiments demonstrating that people tend to remember behaving in ways that are consistent with their fairness. The authors explain this as a motivated reasoning account, something that looks very consistent with identity-based versions of cognitive dissonance theory (e.g., Stone & Cooper, 2001, *Journal of Experimental Social Psychology*), but the details of this mechanism are very underdeveloped and never directly tested. I do not doubt the results of the experiments or that people misremember their past behavior in a manner that is consistent with their fairness beliefs, but there is no evidence in this paper that this result is produced by motivated reasoning. Instead, a much simpler account is that people, at least in part, reconstruct their past behavior based on an inference from their fairness beliefs. That is, left with a fuzzy memory about what they actually did (on average), people's memory for their behavior is guided partly by what they thought was fair. This is especially likely given that the authors set up a procedure where it was actually somewhat hard to remember what you actually did. All of the results in the manuscript are every bit as consistent with this account as they are with the motivated reasoning account offered by the authors, and is a much simpler explanation for the observed behavior.

One key challenge with the current paper is that they do not specify how they think the process of motivated reasoning is actually unfolding in enough detail so that they could test it directly. As I thought about it more, the motivated reasoning account seemed less and less plausible. The authors propose that people misremember their past behavior in order to ward off negative emotions and threats to their moral self-image. But how does this actually work? Are people first aware that there is an inconsistency between their moral self-view (measured as a sense of fairness) and what they actually did (measured as their response in the dictator game)? If so, then at some point people must be accurate in their memory, sense a threat to their self-view, and then somehow deceive themselves into misremembering their past behavior (something that again is easier to presume in others than to actually demonstrate; Gur & Sakheim, 1979). How this latter part happens is never explained. This calls to mind for me the famous New Yorker cartoon by Sidney Harris of two scientists standing in front of a blackboard full of equations, in the middle of which is, "then a miracle occurs."

Again, much simpler than this unclear and underspecified process is the possibility that people's belief about fairness is biasing their memory recall in purely cognitive fashion. If people tend to behave a little more selfishly than what they think is fair, in varying degrees, then the data observed in this experiment is exactly what you'd expect to see. Those who have an especially large gap between their beliefs and their behavior will recall their behavior in a direction biased by their fairness beliefs, as was demonstrated multiple times. But people's moral self-views would not vary based on their actual behavior, nor would their affect, exactly as was found in Study 4.

Without carefully specifying, and then directly testing, the motivated reasoning account described by the authors, I'm afraid this paper is misleading in what it claims to provide to the empirical literature. It is not enough just to argue against alternative accounts for this paper to be publishable. The paper must specify what goal reasoning is trying to achieve, and then demonstrate that reasoning is actually satisfying that goal or at least being actively guided by it. The authors in several places suggest that people are misremembering their past behavior to diminish the threat to their identity, in which case there must be some threat experienced in between the time when people make their choices in the dictator game and then when they recall their behavior. I think a paper that actually specifies a motivated reasoning account of these experiments in detail, and provides positive evidence of it, would be very interesting and worth publishing. This paper is not close to that contribution at this point.

Reviewer #3 (Remarks to the Author):

The authors report three behavioral experiments investigating the accuracy of participants' memory for their own generosity. In a series of Dictator Games, participants decided what percentage of a starting endowment to give to an anonymous recipient (generosity) and were later asked to report the average percentage they had shared across decisions (memory). Prior to the DG, participants had been asked to report the maximum amount that would be fair for the Dictator to keep (fairness standards). Participants on average reported being more generous than they actually were, but this distortion was evident only among those who had violated their own self-reported standards of fairness. The authors conclude that mis-remembering the extent of one's own generosity enables people to simultaneously satisfy their competing desires to act in their own self-interest and to maintain a moral self-image.

Overall, this is a nice set of studies that has the potential to advance the literature on motivated misremembering, primarily by (i) moving beyond general measures (self-reported vividness of a memory) to specific and verifiable ones (how much one actually gave to the other person) and (ii) moving beyond measures of participants' memory for moral rules, as in past studies, to their memory for their behavior itself. That is, although the idea of motivational effects on memory in the moral domain is not new, the present paper documents a new class of such effects. As a result, the findings are likely to be of particular interest to those studying motivated cognition, especially in moral contexts.

The conclusions of this paper rest on three main foundations: participants' behavior, their moral standards, and their memory. My main concerns are about the ability of the reported studies to capture the latter two (moral standards and memory), in part due to potential confounds. I also have some clarification questions about the methods and relation to past work.

Main points

1. Memory distortion vs. memory difficulty.

Given the information presented, a potentially damaging alternative explanation is that the observed differences between self-reported standards and behavior might have arisen due to something about the structure of the decision sets of violators, and/or the math involved in averaging them, rather than something about the psychology of memory following a moral violation. That is, might the five decisions by the violators have had characteristics that made them intrinsically less memorable – or intrinsically more difficult to integrate?

a. To help address the possibility that these results might be due to something about the structure of the decision sets, it would be helpful if the authors considered that structure, and particularly the structure of the sets of violators relative to non-violators. E.g., given that violators were more variable in their responses, might it be the case that determining the average was more difficult for them? Did violators tend to violate on every decision, on a majority, or on a minority of them? By how much? Were people categorized as violators just more careless, or paying less attention, such that they might have inadvertently given less and not remembered it?

b. If it is not possible to rule out set-related concerns otherwise, two approaches could help to do so empirically (albeit in additional studies, which I would suggest only if the concern cannot be addressed adequately by other means):

One approach would be to show a new group of "observer" participants the five decisions of various participants from the current studies (one at a time, in the order in which they were made by the current studies' participants) and then later ask these observers the average percentage given. (That is, replicate the paradigm of the current studies but have participants see each decision as it was made rather than make each decision themselves.) If observers also show more memory distortions for the sets of decisions made by the ultimate violators than by the ultimate upholders (by overestimating violators' generosity), this would point to an alternative explanation, potentially having to do with structure of those sets. On the other hand, to the extent that observers did not show the memory bias shown by violators, this would help to rule out the possibility that there's something about the structure of those sets that affects how participants integrate the choices, thereby helping to rule in the authors' favored explanation that there's something about the psychology of the norm violation that distorts people's memory. Another would be to use the same methodology as in the current experiments but to

test participants' memory for their five decisions individually.

2. Memory distortion vs. lying.

Given the available information, it seems a bit too plausible that these results might not capture memory distortion at all but rather capture lying. Although the use of incentives helps somewhat, is it not possible that the value of appearing moral (by reporting a higher giving rate) simply outweighed the value of the reward for accuracy? The authors dismiss this possibility very quickly in the discussion, but it deserves more earnest attention. In particular, the three pieces of evidence the authors marshal in defense of the memory distortion account do not seem to speak very strongly against this alternative. Lower memory accuracy is subject to the same explanation as memory distortion; reporting lower confidence seems like exactly what one would expect if participants were trying to appear generous but maintain plausible deniability about having lied about it (but perhaps this one would be strengthened if the authors spelled out the logic of their prediction that fairness violators should be less confident in their memory than fairness upholders); and although I appreciate the authors' efforts in using the socially desirable responding scale, it seems to be a lot to ask of this scale to rule out any role for state motivations to appear generous. This strikes me as a real alternative explanation that should be treated seriously, at least rhetorically if not empirically.

3. Moral standards.

a. Construct validity of moral standards measure. At the beginning of the experiment, participants are asked to report the maximum amount that would be fair for the person in the Dictator role to keep for him- or herself. The authors take this number to indicate participants' moral standards, and they accordingly take participants' decisions to keep more than this amount as a violation of those standards. However, it's not clear that this question necessarily captures participants' moral standards so purely. As a result, there are several alternative reasons why a participant's behavior might diverge from the "fair" amount they reported, and it would be useful if the authors made some efforts to rule some of these out. For example, one possibility is that people have different moral standards for themselves than for other people. On this view, asking participants about what would be fair for an abstract person to do might generate a different response than asking what would be fair for the participants themselves to do. Another possibility is that some participants see this question as a potential opportunity for strategic communication, perhaps thinking that there might be some chance that their answer could be transmitted to the other person, and deliberately exaggerate (or diminish) their estimate of what is fair to give to the recipient in service of self-interest. If, in the veil of ignorance condition, some participants exaggerate and others do the opposite (e.g., to pre-justify lower offers, as the authors note), this could result in a wash overall, despite the different motives driving participants' fairness reports. Of course, these examples are not meant to be exhaustive but just to illustrate the form of the concern. What is important is that in both of these cases, the discrepancy between participants' behavior and their moral standards would arise not because participants violated their moral standards per se but rather because

they didn't actually report the relevant standards in the first place. It would be helpful if the authors could provide additional information that would speak to the validity of this measure as an index of personal moral standards. E.g., what was the correlation between self-reported standards and generosity?

b. Effect of having been asked about moral standards. All participants report what would be a fair amount to share prior to deciding how much to share. The authors then interpret violations of participants' self-reported fairness threshold as violations of participants' moral standards and interpret the resulting memory distortions as enabling a moral self-view despite those violations. Are there reasons to favor this interpretation over the narrower interpretation that the observed results pertain not to consistency with one's moral standards writ large but rather to consistency with what one says? That is, regardless of what one's standards are, to what extent could participants' desire for internal consistency between what they say and what they do explain the observed effects?

Clarifications

4. Decision content. The authors write that participants "were informed that one of their decisions would be randomly selected and implemented, and that the size of the endowment would also be randomly determined." Please clarify whether or not participants knew the size of the endowment when deciding what percentage of it to allocate to the other person or whether they simply allocated percentages of endowments whose size was unknown.

5. Choice presentation. The authors note in the supplement that some choices were presented in a graphical format and others in a numeric format. To clarify what participants actually did in these experiments, more information about what these formats actually were would be useful (perhaps just by showing examples in the supplement itself).

6. Memory measure. The authors write that participants were asked to "indicate what percent of the endowment, on average, they transferred to the recipient."

a. In what format were these percentages indicated? In what increments? Was the format of this scale/deposition the same as the one in which they made the decisions in the first place? These details are important because they speak to the extent to the kind of memory processes that might be involved in participants' assessments of how much they gave on average.

b. This question seems somewhat ambiguous between two different ways that the average percentage might be calculated. For simplicity, let's say there were just two decisions per participant. If on two decisions I gave 10% of \$30 (\$3) and 50% of 10 (\$5), is my average giving across those two decisions 30% (the average of 10% and 50%, setting aside endowment) or is my average giving 20% (the percentage

given of the total amount of money at play, i.e., $(3+5=) 8 / (30+10=) 40$? Either way, is it possible that participants were confused about this?

7. Analysis. In Study 1, the authors justify their use of nonparametric statistics; in Studies 2 and 3, the same approach is used without explanation. Is this just for consistency? Or is there independent justification for using Wilcoxon in Studies 2 and 3?

8. Reporting. The authors write, e.g., “However, we found that participants showed a systematic bias towards self-serving memory errors, such that recalled generosity was significantly greater than actual generosity (wilcoxon.sign-ranked = 1345.5, $p = .021$, $d = .23$, Cliff’s $\delta = .14$).” In this and other studies, what were the mean percentages recalled vs. actual?

9. Situation in the context of relevant literature.

a. Although past research on the accuracy of memory for moral violations is presented as “mixed”, there is little discussion of how the current work might be reconciled with those past findings.

b. The authors nicely distinguish vividness and accuracy as two different aspects of memory, and they note that work on vividness “leaves open the question of whether memory accuracy may serve as another mechanism through which individuals can act selfishly and still feel moral”. It would be worth mentioning that there has been other work on accuracy, even if it doesn’t focus on the accuracy of the behavior, e.g., Shu et al’s study in which participants forget the content of a code of conduct they signed, which also concerns specific details (details about moral standards).

Typos

“the mean percent participant’s recalled giving” (Fig 1 caption) - > participants

“we next assessed whether participant’s objective level of generosity” - > participants’

“the average time participant’s took” -> participants

Response to Reviewer 1

We are glad that you thought our paper “*makes a valuable contribution to both the morality and the self-deception literatures*” and “*deserves publication in a highly prestigious outlet.*” Your comments and suggestions were very helpful for revising and improving our manuscript. We have rewritten and reinterpreted our findings on the basis of your suggestions. Below we describe in detail how we responded to all key points mentioned in your report (text written in blue color).

R1.1: Median splits

You note that we report analyses using median splits, which some researchers have questioned.

We agree median splits are not the most appropriate method for analysis. Median splits were indeed used simply for illustrative purposes, and in each experiment, we support these analyses with continuous (regression-based) analyses which produce the same conclusions.

R1.2: Clarifying details of the modified dictator games

You asked how participants experienced playing the modified dictator games in our experiments.

We have now clarified these details in the method sections for Experiments 1-3 on p. 9, p. 15, and p. 20, respectively:

“Decisions were made consecutively on separate screens. After the participant selected and confirmed their response to a decision, the screen advanced to the next decision after a 1-second delay.”

“As in Experiment 1, decisions were made consecutively on separate screens. After the participant selected and confirmed their response to a decision, the screen advanced to the next decision after a 1-second delay.”

“Like Experiments 1 and 2, decisions were made consecutively on separate screens. Before each decision screen, participants saw an intermediary screen for 3 seconds that said “Round # will now begin.”

R1.3: Why would participants make different allocations across games?

You asked why participants would make different allocations in different games, rather than making the same allocation in all five games. You requested data on what proportion

of participants made different vs. uniform allocations across games, suggesting that variability in allocations is relevant to the memory data.

Indeed, a considerable number of participants in each experiment made non-identical decisions—such that the standard deviation of these individuals’ decisions (or *decision variance*) was non-zero (Experiment 1: N = 69/109; Experiment 2: N = 129/234; Experiment 3: N = 223/604). Although the question of why people exhibited choice variability is not a primary focus of this manuscript, we have added a brief explanation and references to the Supplementary Results section (p. 3):

“**Note on choice variability.** Above we show that our findings remain consistent when only including those who made dynamic decisions. Of course, one question that is relevant to our research is why participants would make different decisions across the five different rounds. We offer three explanations for why we expected this to occur. First, people tend to have uncertainty about their preferences, which would manifest through decision variance (Moutoussis, Dolan, & Dayan, 2016). Moreover, even when one’s preferences are certain, the cognitive and neural mechanisms that translate preferences into choices tend to be somewhat noisy (Fehr & Rangel, 2011). Finally, each allocation decision could be influenced to some extent by thoughts and feelings experienced after one’s previous allocation decision (e.g., satisfaction felt after making a fair decision may make people less moral on the next trial, or guilt experienced after an unfair decision may lead people to make a more generous distribution on the next trial (Gneezy, Imas, & Madarász, 2014; Mullen & Monin, 2016). As such, we expected people to exhibit variability even when facing the same choice repeatedly.”

Since it is much easier to recall five identical decisions than five different ones, we controlled for decision variance in several ways to ensure that our results cannot be explained simply by differences between violators and upholders in their level of decision variance. First, we controlled for choice variability in all regression models reported across experiments. Second, we re-ran all key analyses for our three experiments while *excluding* those who had zero decision variance (i.e., “static deciders”). Crucially, all key findings remain the same in these analyses (see below). We have highlighted this information for readers in the main text on p. 11:

“Since decision variance could directly impact the memorability of one’s decisions, we also confirmed that all key findings remain consistent when including only those participants who made non-identical allocation decisions (See SI Appendix).”

“Experiment 1 results (with static deciders excluded)

	test statistic	p	d	δ
Static deciders excluded (N=69)				
1. recalled vs. actual giving (all)	1268.5	.020	.28	.22
2. recalled vs. actual giving (violators)	307.5	.004	.43	.34
3. recalled vs. actual giving (upholders)	342	.45	.17	.11
4. memory errors (violators vs. upholders)	506.5	.31	.23	.14
5. memory accuracy (violators vs. upholders)	739.5	.076	.24	.25

Experiment 2 results (with static deciders excluded)

	test statistic	p	d	δ
Static deciders excluded (N=129)				
1. recalled vs. actual giving (all)	3742	.014	.26	.16
2. recalled vs. actual giving (violators)	1289.5	< .001	.54	.38
3. recalled vs. actual giving (upholders)	595	.39	.02	-.06
4. memory errors (violators vs. upholders)	1321	< .001	.58	.36
5. memory accuracy (violators vs. upholders)	1748.5	.11	.26	.16
6. recalled vs. actual giving (unkind upholders)	118	.79	.03	<.001
7. recalled vs. actual giving (kind upholders)	190.5	.39	.05	-.11
8. memory errors (U vs K upholders)	479	.61	.08	.08

Experiment 3 results (with static deciders excluded)

	test statistic	p	d	δ
Static deciders excluded (N=223)				
1. recalled vs. actual giving (all)	6778	.010	.21	.11
2. recalled vs. actual giving (violators)	1473	< .001	.39	.25
3. recalled vs. actual giving (upholders)	1886	.80	.05	.02
4. memory errors (violators vs. upholders)	4721.5	.008	.39	.21
5. memory accuracy (violators vs. upholders)	5016.5	.045	.30	.16
6. recalled vs. actual giving (unkind upholders)	400.5	.46	.19	.02
7. recalled vs. actual giving (kind upholders)	557	.75	.07	.03
8. memory errors (U vs K upholders)	2197.5	.75	.26	.03

R1.4: Did participants help themselves forget?

You asked whether participants strategically varied their responses to help themselves forget acting immorally.

This is an interesting possibility, and is consistent with our data showing that people who had higher decision variance tend to exhibit lower memory accuracy. Moreover, it seems intuitive that if I am selfish on 3/5 decisions, transferring 8% three times (mean of 8%) will be easier for me to recall than transferring 2%, 8%, and 14% (also a mean of 8%). In the latter case, I can also cherry pick remembering the *least selfish* of my selfish allocations (14%). While our experiments were not designed to test this question, we think it is an exciting future direction for this work. We now discuss this possibility in the discussion section on page 28:

“This raises at least one key direction for future work: since misremembering was more likely to occur when people *varied* in their decisions, it could be strategic to vary the specific amount one gives each time to facilitate motivated misremembering.”

R1.5: Ruling out lying as an alternative explanation

You suggested an alternative account of our findings: participants in fact remembered being unfair, then simply lied about it when reporting how much they gave in the memory quiz. You proposed that the three points we raise in our Discussion to rule out a lying account—namely, that violators have (i) lower memory accuracy, (ii) lower memory confidence, and (iii) no difference in social desirability—were not fully convincing. In particular, you suggested that our memory confidence finding may instead be consistent with participants lying and strategically reporting lower confidence in case they are found out later.

We appreciate your skepticism, and agree that some of the points we previously made to rule out a lying account were not completely convincing. Although we cannot fully exclude the possibility that participants remembered being unfair and lied about it on the memory quiz, we believe that it is highly unlikely. This is because our experiments were designed such that participants have no incentive to lie. In the memory recall stage, we informed participants that they could earn a bonus depending on how close their recalled donations were to their true donations, and participants then reported their recalled donations. It was made clear to participants that there would be no future interaction with any partners. Thus, in this setup, the only “audience member” for a potential lie is the experimenter. However, in order to award the bonus, it is necessary for the experimenter to compare the recalled donations to the true donations. Because the experimenter knows the true donations, it is not possible to successfully lie to the experimenter. Moreover, knowing that an experimenter can verify one’s response has been shown to discourage lying (Gneezy et al., 2018). Thus, it seems highly unlikely that our participants deliberately over-reported their donations to the experimenter because the value of appearing moral to the experimenter outweighed the value of the accuracy bonus. We have now highlighted these points in our Methods (p. 8) to emphasize how our design makes lying unlikely due to experimenter needing to compare recalled versus actual generosity:

“Finally, participants were presented with a surprise incentivized memory test in which they were asked to recall their previous decisions and indicate what percent of the endowment, on average, they transferred to the recipient. To motivate participants to recall their decisions accurately, they were informed that they would receive an additional CHF 5 if their response was within 10% of their actual average transfer. This setup provides participants with no incentive to consciously lie. At the recall phase, participants knew that there would be no future interactions with any partners. They also knew that their reported memories were accessible by the research team—who also required access to the participants true decisions in order to determine their accuracy bonus. Such knowledge is known to reduce lying (Gneezy et al., 2018).”

In our original submission, we reported that violators, in addition to being less accurate in their memories, were less confident. Both you and Reviewer 3 raised the possibility that it would be strategic to report a “lack of confidence” in one’s memory when consciously lying to others about one’s generosity, as it offers plausible deniability to violators in case

they are later “found out”. We do not deny that this (fascinating) phenomenon likely occurs in social life, and we cannot fully exclude the possibility this accounts for some of our confidence findings. However, in light of the fact that participants had no incentive to lie in our experiments, we think it is more parsimonious and more probable that violators’ lower confidence ratings reflected *genuine* difficulty in recalling their decisions, rather than a strategy for maintaining plausible deniability for active deception. To further support this view with empirical evidence, we tested whether confidence ratings were lower among violators who made *self-serving* versus *self-defeating* memory errors. We have added this analysis to the Supplemental Material:

“Experiment 3: ruling out a “confidence ratings as plausible deniability” account. Reporting lower memory confidence could reflect retrieval difficulty, but it could also reflect a strategy for maintaining plausible deniability if one were lying about their level of generosity. To address this possibility, we tested whether confidence ratings were lower among violators who made self-serving versus self-defeating memory errors. A lying account would predict that confidence ratings should be lower specifically among those who made self-serving memory errors (i.e., those predicted to have a need to maintain plausible deniability). However, we found no significant difference in confidence ratings between violators who made self-serving memory errors ($M = 4.95$, $SD = 1.75$) versus self-defeating memory errors ($M = 4.35$, $SD = 1.60$; $W = 324.5$, $p = .14$, $d = .35$, $\delta = .23$).”

Finally, we have modified our Discussion section (p. 24) to integrate the above points:

“Specifically, these findings suggest that those who violate (as opposed to uphold) their personal standards tend to misremember the extent of their selfishness. An alternative account would hold that these individuals were aware of their true level of generosity at recall, yet were willing to pay a cost to claim having been more generous. This explanation is consistent with prior work (Mazar et al., 2008), and we cannot fully exclude the possibility it accounts for our data here. However, we think this account is highly unlikely to explain our results. In addition to incentivizing accurate recall, we also removed incentives to consciously lie. Participants knew there would be no future interaction with any partners. Thus, the only “audience” for a potential lie was the experimenter. Since participants also knew that the experimenter needed to verify their reported memories against their true decisions in order to award bonus payments, there was no reason for them to lie (Gneezy et al., 2018). Additionally, we found little to no effect of trait social desirability on peoples’ reported memories. Together, these points suggest that people were actually misremembering their decisions, rather than consciously lying about them.”

Response to Reviewer 2

We are glad you saw potential in our paper, suggesting it would be “*very interesting and worth publishing*” if we could specify a “motivated” account of our findings and provide positive evidence for it. Your comments and suggestions encouraged us to make a number of changes to the manuscript. Most notably, we added several new analyses that can adjudicate between motivated and rational inference accounts of our findings. Below we describe in detail how we responded to all key points mentioned in your report (text written in blue color).

R2.1: Situating our work within the broader literature on motivated reasoning

You highlighted an important historical debate in the field that could directly inform our theoretical framework (e.g., Festinger & Carlsmith, 1959; Bem, 1972; Kruger, 1999; Chambers & Windschitl, 2004; Stone & Cooper, 2001). Namely, you suggested that *motivated reasoning accounts* of inconsistencies between attitudes and behavior have been challenged by simpler, *rational inference* accounts of the same phenomena. You requested that we consider this literature in clearly articulating and testing a “motivated” account for our data.

Aronson (1992) once wrote “social psychology has a long history and a very short memory”. We can imagine you were reminded as such reading our paper, as we failed to mention many relevant works from the history of our field. We appreciate your encouragement to integrate them into our paper, which we have revised considerably in line with your suggestions. In particular, we have reframed our introduction to highlight rational inference versus motivated accounts of misremembering (p. 6).

“A rich, long-standing debate in social psychology concerns whether self-serving biases require motivation (Bem, 1972; Chambers & Windschitl, 2004). For instance, many seemingly ‘motivated’ social comparative biases (e.g., above-average effects) can arise from purely rational inference processes (Chambers & Windschitl, 2004; Fiske & Taylor, 1984; Kruger, 1999; Tversky & Kahneman, 1981). Although the moral domain is one where motivated effects are well-supported (Kappes & Crockett, 2016; Mazar, Amir, & Ariely, 2008), it is nonetheless crucial to rule out the possibility that rational inference processes may produce misremembering that only seems morally motivated. For instance, since people tend to be fair across many social situations, it is reasonable for them to infer they behaved fairly in the past. That is, if memory for recent behavior is weak or incomplete, people may tend to rely on knowledge of what they normally do to inform recall.”

We also add mechanistic precision to our hypotheses by distinguishing between reasoning processes—a major focus of past work—from memory processes—an area that has received less attention. To be clear: we do not see our work as adding to an extensive literature on motivated reasoning. Rather, our goal is to document a new class of motivated effects on *memory* by showing that people form *less accurate memories* of

behaviors that fall short of their personal standards. In articulating this hypothesis, we highlight when motivated *reasoning* is likely to occur (e.g., when justifying one’s actions post-hoc), and when we think motivated *misremembering* is most likely to occur. These changes are reflected in our Introduction (p. 4-5):

“How can people preserve their moral self-image while simultaneously acting selfishly? Social scientists have often credited our ability to engage in *motivated reasoning* (Kunda, 1990)—that is, we skillfully justify immoral acts to ourselves before or after the events unfold (Mazar et al., 2008; Shalvi, Gino, Barkan, & Ayal, 2015; Walster, Berscheid, & Walster, 1973). This feat is accomplished in a number of ways. For one, people tend to strike a justifiable balance between self-interest and their moral values – for instance, lying just enough to profit financially, but not so much as to harm their moral self-image (Mazar et al., 2008). In addition, people psychologically *distance* themselves from their unethical actions – attributing past misdeeds to situational pressures (Malle, 2006), or having been a “different person” at the time (Ross & Wilson, 2003; Stanley, Henne, Iyengar, Sinnott-Armstrong, & De Brigard, 2017). Moreover, people exploit *uncertainty* – behaving more selfishly when the consequences for others are ambiguous (Bénabou & Tirole, 2011; Exley, 2015), and avoiding information about how their actions may have harmed others (Dana, Weber, & Kuang, 2007; Feiler, 2014; Larson & Capra, 2009).

Another possibility that has received less attention is that our desire to believe we are moral may permeate our memories. When people’s actions fall short of their personal standards, they might actually remember acting in line with those standards. Failing to remember past immoral actions would *pre-empt* the need to engage in motivated reasoning, because behavior that falls short of one’s personal standards is simply forgotten. This idea coheres with evidence that people are able to “suppress” awareness of unwanted memories at both encoding and retrieval (Anderson & Hanslmayr, 2014; Hu, Bergström, Bodenhausen, & Rosenfeld, 2015), as well as evidence that memories for dishonest behavior (e.g., cheating) are less *vivid* than memories of honest actions (Kouchaki & Gino, 2016). Such work suggests that the subjective experience (i.e., vividness) of recalling unethical actions may be impaired. However, it is less clear whether the objective details (i.e., accuracy) of such memories are affected. Thus far, initial work on this question has yielded mixed results (Kouchaki & Gino, 2016; Saucet & Villevall, 2018; Stanley, Yang, & Brigard, 2018), or has focused on memory for abstract events (e.g., hypothetical situations; Kouchaki & Gino, 2016) and concepts (e.g., rules; Shu & Gino, 2012).

R2.2: Can we rule out a rational inference account for our findings?

You asked whether we can provide evidence that misremembering in our experiments is due to a *motivated* process. You suggested that our findings could be equally consistent with a rational inference account: left with a fuzzy memory about what they actually did, people’s memory for their behavior is guided partly by what they thought was fair.

We appreciate your thoughtful challenge as to whether our observation of misremembering reflects a motivated process, or can instead be explained by a purely rational inference account. Below, we provide additional evidence that is hard to reconcile with a rational inference account and fully consistent with a motivated misremembering account.

In our experiments, motivated versus rational inference accounts make different predictions about the nature of misremembering. A rational inference account states that those with a gap between their beliefs and their behavior will recall their behavior in a direction biased by their fairness beliefs. In other words, when one's actions *deviate* from one's norm, the memory of those actions will tend to move *toward* one's norm. This account predicts that in our experiment, people who gave *less* than their norm (violators) should misremember in the direction of their norm, as we found. However, this account also predicts that individuals who gave *more* than their norm ("exceeders") should misremember in the direction of their norm. That is, violators should recall giving more than they actually did, and exceeders should recall giving less than they actually did.

In contrast, if people are *motivated* to maintain a positive self-image, violators should be motivated to misremember giving more than they actually did (to reduce the threat to their moral self-image), while exceeders should be motivated to recall their actions accurately (given that accuracy is incentivized, and their decisions pose no threat to their self-image). Thus, a rational inference account predicts a *symmetry* in misremembering among violators *and* exceeders, whereas a motivated account predicts an *asymmetry* in misremembering between violators and exceeders, such that *only* violators misremember.

To test these competing predictions, we created a new group of "exceeders" which only included those who gave more than what they personally believed was fair (Experiment 2, $N = 69$; Experiment 3, $N = 207$), and excluded those who simply gave the exact amount that they said was fair. Crucially, one-sample signed rank tests showed no evidence for misremembering 'toward the norm' in exceeders (Experiment 2, $V = 538$, $p = .34$, $d = .03$, $\delta = -.07$; Experiment 3, $V = 1553$, $p = .80$, $d = .06$, $\delta = 0.02$). Instead, exceeders, unlike violators, appeared to be extremely accurate in recalling their decisions. These results are consistent with a motivated account, and inconsistent with a rational inference account. We have illustrated this point for readers and included these analyses in the Results sections for Experiments 2 and 3 (p. 17-18; p. 22):

Next, we sought evidence that the misremembering we observe here is in fact *motivated* (i.e., driven by a desire to preserve one's moral self-image) rather than based on a *rational inference* (i.e., reconstructed from past behavior based on an inference from their fairness beliefs). If participants, when faced with difficulty recalling their true decisions, simply reconstruct those decisions based on an inference from what they believe is fair, we would predict that those who gave less than their personal standard (i.e., violators) would tend to recall giving *more* than they actually did, while those who gave more than their personal standard (i.e., exceeders) would tend to recall giving *less* than they actually did. However, if misremembering is in fact motivated, we would only expect misremembering to occur among violators, whereas exceeders should tend to recall their decisions accurately. To test these possibilities, we examined memory errors in exceeders ($N = 60$, average generosity = 30.3%)—a group that showed a similar degree of norm deviation and decision variance (See *SI Appendix*). In support of our hypothesis that misremembering is motivated, we found no evidence of misremembering towards the norm in exceeders ($V = 538$, $p = .34$, $d = .03$, $\delta = -.07$), and violators misremembered to a significantly greater degree than exceeders ($W = 1209$, $p < .001$, $d = .58$, $\delta = .37$).

Nevertheless, even if exceders show no bias *on average*, a rational inference account would still find partial support if exceders were more likely to make errors in the direction of their norm (i.e., *norm-directed memory errors*) than in the opposite direction. To address this question, we compared the frequency of errors made in each direction by exceders and violators. For this analysis, we were limited to a smaller sample as we were only able to analyze data from those violators (N = 56) and exceders (N = 50) who in fact made memory errors. A binomial test indicated that, whereas the proportion of violators who made norm-directed memory errors (71.4%) was greater than chance ($p = 0.002$, 95% CI [.58, .83]), the proportion of exceders who made norm-directed memory errors (54%) was no different than that expected by chance ($p = .67$, 95% CI [.39, .68]). These results are also inconsistent with a rational inference account, but consistent with a motivated account of misremembering.”

And when discussing the results of Experiment 3, we added the following paragraph:

“We next examined memory errors in participants who gave *more* than what they indicated was fair (*exceders*, N = 123, average generosity = 34.5%). In support of our hypothesis that misremembering is motivated, we found no evidence of norm-directed memory errors in exceders ($V = 1553$, $p = .80$, $d = .06$, $\delta = 0.02$), and violators misremembered to a significantly greater degree than exceders ($W = 4301$, $p = .010$, $d = .39$, $\delta = .21$)—supporting the idea that misremembering is indeed motivated.

To provide further support for this view, we compared the frequency of errors made in the same (versus opposite) direction of their norm by violators (N = 62) and exceders (N = 77) who in fact made memory errors. In support of a motivated account of misremembering, a binomial test indicated that, whereas the proportion of violators who made norm-directed memory errors (67.7%) was greater than chance ($p = 0.007$, 95% CI [.55, .79]), the proportion of exceders who made norm-directed memory errors (48.1%) was no different than that expected by chance ($p = .82$, 95% CI [.37, .60]).”

Together, these findings provide evidence that the phenomenon observed in our experiments is *hard to reconcile* with a rational inference account while it is fully consistent with a motivated account of misremembering. We are very grateful that you encouraged us to discriminate between these two competing hypotheses by performing additional analyses. We believe that our paper has benefitted greatly by discussing and addressing this competing theoretical account.

R2.3: Specifying and testing a motivated account of our experiments

You suggested that we specify and test a motivated account of our experiments, including a description of what goal this process is trying to satisfy, and a demonstration that misremembering is actually satisfying that goal or at least being actively guided by it. Specifically, you suggested that “*a paper that actually specifies a motivated reasoning account of these experiments in detail, and provides positive evidence of it, would be very interesting and worth publishing.*”

These points are well taken. Before we address them, we think it is important to further distinguish *motivated reasoning* processes from *motivated memory* processes. Our focus on memory goes beyond prior work on motivated reasoning (e.g., self-perception and cognitive dissonance models) by positing a different set of mechanisms through which

dissonance can be resolved. For instance, Stone and Cooper's (2001) model assumes that people either self-affirm or self-justify after a discrepant action to relieve their dissonance. We show that there is another path beyond these reasoning processes to reduce dissonance: simply forgetting that an action was ever discrepant to begin with. That is, we suggest that people's desire to reduce dissonance can shape memories of concrete actions. This is categorically different than reducing dissonance by changing attitudes toward abstract entities (e.g., support for a policy, or beliefs about 'the average person'). By highlighting a novel psychological path by which people resolve dissonance, we believe this work constitutes an important contribution to the literature.

Below, we address your specific points by making the case that (i) we specify and test a motivated account for our findings, and (ii) we describe this account with a reasonable level of detail, given the scope of the current research.

We specify and test a motivated account for our findings.

Based on your suggestions, we have revised our introduction to clearly specify a motivated account of misremembering, as well as a description of the goal this process is hypothesized to satisfy:

“When behaving unfairly (e.g., giving a stingy tip), people may misremember the extent of their selfishness, thus preserving the view that they treat others fairly. To our knowledge, it is unknown whether people engage in such *motivated misremembering* of selfish actions – that is, remembering having been more generous than they truly were.

Here, we tested this possibility. Across three experiments, we investigated whether selfish decisions are misremembered as objectively more generous. Crucially, we devised experiments in which motivated reasoning processes should have a minimal influence—recalling a recently performed action for which one's standard of fairness is explicitly known. If one engages in motivated reasoning before or after engaging in unethical behavior, there should be no reason to misremember a justified action. If people instead tend to engage in motivated misremembering, such biases should be revealed at recall. We predicted that motivated misremembering would be evident among more selfish individuals, and particularly those whose actions violate their own standard of fairness.

Another aim of this work was to investigate whether misremembering is in fact *motivated*. A rich, long-standing debate in social psychology concerns whether self-serving biases require motivation (Bem, 1972; Chambers & Windschitl, 2004). For instance, many seemingly 'motivated' social comparative biases (e.g., above-average effects) can arise from purely rational inference processes (Chambers & Windschitl, 2004; Fiske & Taylor, 1984; Kruger, 1999; Tversky & Kahneman, 1981). Although the moral domain is one where motivated effects are well-supported (Kappes & Crockett, 2016; Mazar et al., 2008), it is nonetheless crucial to rule out the possibility that rational inference processes may produce misremembering that only seems morally motivated. For instance, since people tend to be fair across many social situations, it is reasonable for them to infer they behaved fairly in the past. That is, if memory for recent behavior is weak or incomplete, people may tend to rely on knowledge of what they normally do to inform recall.”

In addition to ruling out the possibility that misremembering in our experiments is driven by purely rational inference processes (see our response to your previous query above), we provide positive evidence for a motivated account. Crucially, we show across our experiments that only when people *violate* their personal standards do they misremember their past behavior. This is precisely the type of scenario in which the self-standard model (Stone & Cooper, 2001) would predict cognitive dissonance to occur. A personal standard was made situationally accessible (i.e., personal fairness standards), and people who freely chose to violate this standard showed memory errors that systematically reduced the discrepancy between their standards and their behavior. These errors occurred despite being incentivized to recall their actions accurately.

Because one goal of motivated misremembering is to provide emotional benefits in the wake of behavior that falls short of one's personal standards, a motivated account *specifically* predicts that emotional benefits should occur for (i) violators who (ii) misremember in the direction of their norm (versus the non-normative direction). This is precisely what we found in Experiment 3. Violators who made self-serving memory errors reported greater positive affect than those who made self-defeating memory errors, whereas upholders derived no emotional benefits from self-serving errors compared to self-defeating errors:

“Since we specifically predicted that positive affect should be improved among violators who made self-serving memory errors, we tested whether positive affect differed specifically when violators made self-serving (versus self-defeating) memory errors. For violators, making self-serving (vs. self-defeating) memory errors significantly impacted positive affect ($W = 301.5, p = .031, d = .56, \delta = .33$), such that violators who exhibited self-serving memory errors were happier than those who did not (Fig. 2B). However, no such shift in positive affect was observed among upholders ($W = 1203.5, p = .68, d = .08, \delta = .05$).”

Fig 2 (B) In addition, violators who made self-serving memory errors reported greater positive affect than those who made self-defeating memory errors. Crucially, this difference was not observed among upholders. Error bars reflect SEMs. This figure is re-visualized with all raw data points in the *SI Appendix* (Fig. S3).

These results favor a motivated account over a rational inference account by virtue of their specificity: they support the idea that misremembering is serving the *goal* of reducing psychological discomfort associated with a threatened moral self-image.

We specify our motivated misremembering hypothesis with a reasonable level of detail.

Your review points out that it would be interesting to identify the specific memory process that helps people misremember their selfish decisions. In fact, it is possible that a number of processes could play a role, as ample evidence suggests that motivation can impact numerous memory processes (Amodio & Ratner, 2011; Anderson & Hanslmayr, 2014; Crowder, 2014). However, exciting as we think it would be to tease apart specific memory mechanisms, this is well beyond the scope of the current paper, which simply aims to highlight the motivated nature of memory processes. In our revised Discussion, we have suggested some future directions that may allow one to take a closer look at the memory processes at work in our research (pg 28-29):

“If we accept the possibility that people can misremember their misdeeds, what cognitive and neural mechanisms might facilitate this effect? Motivated misremembering could plausibly emerge at any stage of memory (encoding, consolidation, or retrieval) (Anderson & Hanslmayr, 2014). Below we propose two possibilities, though it is important to note that numerous memory mechanisms could plausibly contribute to misremembering in our experiments (Amodio, 2019; Anderson & Hanslmayr, 2014; Crowder, 2014).

One possibility is that selfish individuals misremember their decisions via mechanisms that unfold at encoding. Memory encoding is chiefly determined by attention (Chun & Turk-Browne, 2007), which in turn is guided by our goals (Al-Aidroos, Said, & Turk-Browne, 2012). Upon first becoming aware of the intention to make a selfish decision, individuals may attend less to episodic details, and strategically downregulate the depth of processing of such episodes via inhibitory control processes (Anderson & Hanslmayr, 2014). As a consequence, violators may misremember at recall due to their most selfish decisions being less available in memory (Gordon, Franklin, & Beck, 2005; Johnson, 2006)—that is, selfish decisions are less richly encoded, and thus more generous decisions loom larger at recall.

Another possibility is that selfish decisions are accurately stored in memory, but individuals later misremember due to retrieval-related processes (Anderson & Hanslmayr, 2014; Kappes & Crockett, 2016). To this end, individuals might suppress retrieval of their more selfish decisions. Engaging in this form of inhibitory control over memory—*suppression-induced forgetting*—would aid individuals not only in ensuring that selfish decisions do not enter conscious awareness, but also in reducing their ability to later recall these decisions (Anderson & Green, 2001; Gagnepain, Henson, & Anderson, 2014). Such findings suggest that even if all of one’s decisions were stored with equal fidelity, their more selfish ones may nonetheless become less accessible over time, ultimately leading them to preferentially recall decisions that are more in line with their moral self-view.

Importantly, these mechanisms of misremembering are neurally dissociable (Anderson & Hanslmayr, 2014; Gruber, Ritchey, Wang, Doss, & Ranganath, 2016; Manning et al.,

2016). For instance, an encoding account of misremembering would be more consistent with decreased functional connectivity between midbrain dopaminergic targets and hippocampus immediately following selfish decisions (Gruber et al., 2016). In contrast, a retrieval account of misremembering would predict increased dorsolateral prefrontal activation and reduced hippocampal activation at encoding or retrieval, reflecting top-down control over memory (Benoit & Anderson, 2012; Gagnepain et al., 2014). Teasing apart the influences of such mechanisms may be fruitfully pursued by examining how neural representations of ethical and unethical memories change over time (Bonnici & Maguire, 2018).”

Additional exploratory analyses. Although we felt it was important to make the above points first, we did conduct exploratory analyses, albeit with a very small sample, that test two competing mechanisms raised in our discussion: encoding versus retrieval effects. We based our analyses on the following premises.

If misremembering is due to an encoding failure, this process should unfold during the decision phase of our experiments. Because any transfer that violates one’s standards could plausibly pose a self-image threat, motivated encoding processes might inhibit encoding of choices that violate one’s standards. Thus, a motivated encoding account predicts that we should observe motivated misremembering among *anyone* who violated their norm at least once, even if they upheld their norm *on average* (i.e., were classified as ‘upholders’ in our experiments).

If misremembering is due to a retrieval failure, this process should unfold during the recall phase of our experiments. Since participants were specifically prompted to recall how much they gave on average, it would be particularly threatening to recall past decisions if one’s *average* transfer violated one’s standards. As such, motivated retrieval processes might inhibit retrieval of more selfish choices if one’s past choices, on average, violated one’s personal standards. Thus, a motivated retrieval account predicts that we should observe motivated misremembering only among people who violated their norm *on average* (i.e., those classified as ‘violators’ in our experiments).

To test these two alternatives, we examined the data of only those who violated their norm at least once, but upheld it on average (N = 14, Experiment 2; N = 28, Experiment 3). Our crucial question was whether these individuals exhibited motivated misremembering or not, and one-sample U tests showed no evidence of misremembering in either Experiment 2 ($V = 27.5$, $p = .65$, $d = .15$, $\delta = -.07$) or Experiment 3 ($V = 94$, $p = .98$, $d = .05$, $\delta = -.11$).

These results suggest that even those upholders who violated their norm on some trials tended to have accurate memory. While we cannot draw strong conclusions from these analyses, these findings are more consistent with a retrieval account of misremembering. As such, these findings, although preliminary, may provide guidance to future work that seeks to further tests the mechanisms behind motivated misremembering. Given the preliminary nature of these findings, we would prefer not to report them in the paper, but if you and/or the editor thinks they are worth reporting, we are happy to do so in the supplemental materials.

Response to Reviewer 3

We are glad you think our paper “*has the potential to advance the literature on motivated misremembering*” by documenting “*a new class of such effects*”. We thank the Reviewer for their thoughtful comments and suggestions. We followed your suggestions and conducted a number of additional analyses, all of which support our central hypothesis and help rule out alternative explanations. Below we describe in detail how we responded to all key points mentioned in your report (text written in blue color).

R3.1 Can results be explained by systematic differences in decision sets of violators vs. upholders?

You suggested that the choice patterns of violators versus upholders may lead to these two groups to be recalling choice sets that may systematically differ in memorability. To rule this out, you suggested that we more closely examine the structure of these choices, and proposed analyses and experiments to conduct to test the role of choice structure in memory.

These are great points, and seem especially important when considering those deciders who “matched” their norm (a subset of our upholder group). This group typically made the same allocation decision 5 times, which was the same amount as their reported norm (e.g., they reported that 50% was fair and gave 50% each time). On the other hand, violators typically exhibited greater choice variability. Thus, these two groups were, on average, recalling differently-structured decision sets.

Since it is much easier to recall five identical decisions than five different ones, we controlled for decision variance in several ways to ensure that our results cannot be explained simply by differences between violators and upholders in their level of decision variance.

As mentioned in our response above to **Comment R1.3**, we controlled for choice variability in all regression models reported across experiments. Second, we have re-run all key analyses for our three experiments while *excluding* those who had zero decision variance (i.e., “static deciders”). These additional analyses are now reported in the Supplement (See SI Appendix), where we also report the number of participants who made non-identical decisions in each experiment. Crucially, all key findings remain the same in these analyses (see below).

Experiment 1 results (with static deciders excluded)

	test statistic	p	d	δ
Static deciders excluded (N=69)				
1. recalled vs. actual giving (all)	1268.5	.020	.28	.22
2. recalled vs. actual giving (violators)	307.5	.004	.43	.34
3. recalled vs. actual giving (upholders)	342	.45	.17	.11

4.	memory errors (violators vs. upholders)	506.5	.31	.23	.14
5.	memory accuracy (violators vs. upholders)	739.5	.076	.24	.25

Experiment 2 results (with static deciders excluded)

	test statistic	p	d	δ	
Static deciders excluded (N=129)					
1.	recalled vs. actual giving (all)	3742	.014	.26	.16
2.	recalled vs. actual giving (violators)	1289.5	< .001	.54	.38
3.	recalled vs. actual giving (upholders)	595	.39	.02	-.06
4.	memory errors (violators vs. upholders)	1321	< .001	.58	.36
5.	memory accuracy (violators vs. upholders)	1748.5	.11	.26	.16
6.	recalled vs. actual giving (unkind upholders)	118	.79	.03	<.001
7.	recalled vs. actual giving (kind upholders)	190.5	.39	.05	-.11
8.	memory errors (U vs K upholders)	479	.61	.08	.08

Experiment 3 results (with static deciders excluded)

	test statistic	p	d	δ	
Static deciders excluded (N=223)					
1.	recalled vs. actual giving (all)	6778	.010	.21	.11
2.	recalled vs. actual giving (violators)	1473	< .001	.39	.25
3.	recalled vs. actual giving (upholders)	1886	.80	.05	.02
4.	memory errors (violators vs. upholders)	4721.5	.008	.39	.21
5.	memory accuracy (violators vs. upholders)	5016.5	.045	.30	.16
6.	recalled vs. actual giving (unkind upholders)	400.5	.46	.19	.02
7.	recalled vs. actual giving (kind upholders)	557	.75	.07	.03
8.	memory errors (U vs K upholders)	2197.5	.75	.26	.03

We also have highlighted this information for readers in the main text as well on p. 9:

“Since decision variance could directly impact the memorability of one’s decisions, we also confirmed that all key findings remain consistent when including only those participants who made non-identical allocation decisions (See SI Appendix).”

Another way we addressed this concern is by comparing violators to better a matched group. A number of participants “exceeded” their norm across our experiments, giving more on average than what they said was fair. These individuals not only deviated from their norm to a similar degree on average as violators, but also exhibited a somewhat similar degree of decision variance, as we report in Supplemental Material for Experiments 2 & 3:

“Norm deviation and decision variance in violators versus exceders. Violators and exceders were relatively well-matched in the extent to which they deviated from their norm (violators: $M = 13.31$, $SD = 10.96$; exceders: $M = 10.80$, $SD = 9.95$; $W = 1656$, $p = .19$), and were somewhat similar in the extent to which their decisions varied (violators: $M = 11.91$, $SD = 6.87$; exceders: $M = 8.75$, $SD = 4.80$), however violators varied to a greater degree ($W = 1353$, $p = .004$).”

“Norm deviation and decision variance in violators versus exceders. Violators and exceders were relatively well-matched in the extent to which they deviated from their norm (violators: $M = 18.59$, $SD = 15.99$; exceders: $M = 15.27$, $SD = 12.45$; $W = 4938.5$, $p = .28$), but also in the degree to which their decisions varied (violators: $M = 11.50$, $SD = 7.497$; exceders: $M = 10.79$, $SD = 7.05$; $W = 5113$, $p = .49$).

Since these two groups are relatively well-matched, they should experience similar difficulty in remembering their choices. Yet when we compare these two groups, only violators showed a significant tendency to distort their memories, which suggests that our results cannot simply be explained by differences in the structure of their decision sets affecting the memorability of their decisions. To clarify this point for readers, we have added these analyses to the Results section (p. 17-18; p. 22):

“To test these possibilities, we examined memory errors in exceders ($N = 60$, average generosity = 30.3%)—a group that showed a similar degree of norm deviation and decision variance (See *SI Appendix*). In support of our hypothesis that misremembering is motivated, we found no evidence of misremembering towards the norm in exceders ($V = 538$, $p = .34$, $d = .03$, $\delta = -.07$), and violators misremembered to a significantly greater degree exceders ($W = 1209$, $p < .001$, $d = .58$, $\delta = .37$).”

“We next examined memory errors in participants who gave *more* than what they indicated was fair (*exceeders*, $N = 123$, average generosity = 34.5%). In support of our hypothesis that misremembering is motivated, we found no evidence of a bias towards self-serving memory errors in exceders ($V = 1553$, $p = .80$, $d = .06$, $\delta = 0.02$), and violators misremembered to a significantly greater degree exceders ($W = 4301$, $p = .010$, $d = .39$, $\delta = .21$)—supporting the idea that misremembering is indeed motivated.”

R3.2: Ruling out lying as an alternative explanation

You suggested that an alternative account of our findings is that participants in fact remembered being unfair, then lied about it when reporting how much they gave in the memory quiz. You proposed that the three points we raise in our Discussion to rule out a lying account—namely, that violators have (i) lower memory accuracy, (ii) lower memory confidence, and (iii) no difference in social desirability—were not convincing. In particular, you suggested that our memory confidence finding may instead be consistent with participants lying and strategically reporting lower confidence in case they are found out later. Reviewer 1 had the same concern (See **Comment R1.5**), thus we have reproduced our reply to their concern below.

We appreciate your skepticism, and agree that some of the points we previously made to rule out a lying account were not completely convincing. Although we cannot fully exclude the possibility that participants remembered being unfair and lied about it on the memory quiz, we believe that it is highly unlikely. This is because our experiments were designed such that participants have no incentive to lie. In the memory recall stage, we instructed participants that they could earn a bonus depending on how close their recalled donations were to their true donations, and participants then reported their recalled donations. It was made clear to participants that there would be no future interaction with any partners. Thus, in this setup, the only “audience member” for a potential lie is the experimenter. However, in order to award the bonus, it is necessary for the experimenter to compare the recalled donations to the true donations. Because the experimenter knows the true donations, it is not possible to successfully lie to the experimenter. Moreover, knowing that an experimenter can verify one’s response has been shown to discourage lying (Gneezy et al., 2018). Thus, it seems highly unlikely that our participants deliberately over-reported their donations to the experimenter because the value of appearing moral to the experimenter outweighed the value of the accuracy bonus. We have now highlighted these points in our Methods (p. 8) to emphasize how our design makes lying unlikely due to experimenter needing to compare recalled versus actual generosity:

“Finally, participants were presented with a surprise incentivized memory test in which they were asked to recall their previous decisions and indicate what percent of the endowment, on average, they transferred to the recipient. To motivate participants to recall their decisions accurately, they were informed that they would receive an additional CHF 5 if their response was within 10% of their actual average transfer. This setup provides participants with no incentive to consciously lie. At the recall phase, participants knew that there would be no future interactions with any partners. They also knew that their reported memories were accessible by the research team—who also required access to the participants true decisions in order to determine their accuracy bonus. Such knowledge is known to reduce lying (Gneezy et al., 2018).”

In our original submission, we reported that violators, in addition to being less accurate in their memories, were less confident. Both you and Reviewer 1 raised the possibility that it would be strategic to report a “lack of confidence” in one’s memory when consciously lying to others about one’s generosity, as it offers plausible deniability to violators in case they are later “found out”. We do not deny that this (fascinating) phenomenon likely occurs in social life, and we cannot fully exclude the possibility this accounts for some of our confidence findings. However, in light of the fact that participants had no incentive to lie in our experiments, we think it is more parsimonious and more probable that violators’ lower confidence ratings reflected *genuine* difficulty in recalling their decisions, rather than a strategy for maintaining plausible deniability for active deception. To further support this view with empirical evidence, we tested whether confidence ratings were lower among violators who made *self-serving* versus *self-defeating* memory errors. We have added this analysis to the Supplemental Material:

“Experiment 3: ruling out a “confidence ratings as plausible deniability” account. Reporting lower memory confidence could reflect retrieval difficulty, but it could also reflect a strategy for maintaining plausible deniability if one were lying about their level

generosity. To address this possibility, we tested whether confidence ratings were lower among violators who made self-serving versus self-defeating memory errors. A lying account would predict that confidence ratings should be lower specifically among those who made self-serving memory errors (i.e., those predicted to have a need to maintain plausible deniability). However, we found no significant difference in confidence ratings between violators who made self-serving memory errors ($M = 4.95$, $SD = 1.75$) versus self-defeating memory errors ($M = 4.35$, $SD = 1.60$; $W = 324.5$, $p = .14$, $d = .35$, $\delta = .23$).

Finally, we have modified our Discussion section (p. 24) to integrate the above points:

“Specifically, these findings suggest that those who violate (as opposed to uphold) their personal standards tend to misremember the extent of their selfishness. An alternative account would hold that these individuals were aware of their true level of generosity at recall, yet were willing to pay a cost to claim having been more generous. This explanation is consistent with prior work (Mazar et al., 2008), and we cannot fully exclude the possibility it accounts for our data here. However, we think this account is highly unlikely to explain our results. In addition to incentivizing accurate recall, we also removed incentives to consciously lie. Participants knew there would be no future interaction with any partners. Thus, the only “audience” for a potential lie was the experimenter. Since participants also knew that the experimenter needed to verify their reported memories against their true decisions in order to award bonus payments, there was no reason for them to lie (Gneezy et al., 2018). Additionally, we found little to no effect of trait social desirability on peoples’ reported memories. Together, these points suggest that people were actually misremembering their decisions, rather than consciously lying about them.”

R3.3: Construct validity of personal standards measure

You raised several questions about the validity of our measure of personal fairness standards. You suggested people could have different fairness standards for themselves than for other people, and that the wording of our question may generate a different response than asking what would be fair for the participants themselves to do. You also raised the possibility that some participants saw the question as a potential opportunity for strategic communication, perhaps thinking that there might be some chance that their answer could be transmitted to the other person, and deliberately exaggerate (or diminish) their estimate of what is fair to give to the recipient in service of self-interest. To address these concerns, you requested that we provide additional information that would speak to the validity of this measure as an index of personal fairness standards, such as providing the correlation between self-reported standards and generosity.

We agree that these are important concerns, which is why in Experiment 2 we asked people what they personally believed would be a fair amount for the Decider to give, both *after* they already knew they were the Decider (in the full knowledge condition), and *before* they knew they were the Decider (in the veil of ignorance condition). If participants have different standards for themselves versus an abstract ‘other’, then one

would expect their reported standard of fairness to be different in the full knowledge condition (where they *knew* they would be the Decider). We did share your intuition that participants would diminish their estimate of what is fair to give when they knew they were the decider (out of self-interest). However, we found no difference between these conditions ($W_{\text{wilcoxon.rank-sum}} = 6198, p = .20$; as previously reported in the Supplemental Material), which supports the idea that this measure elicited people's personal fairness standards in either case.

Another possibility is that some participants in the full knowledge condition tended to deviate *downward* by pre-justifying keeping more for themselves, and others in the full knowledge condition tending to deviate *upward*—reporting that a split closer to 50/50 is fair (as a way to self-signal their morality). This account would predict that participants in the full knowledge condition should report similar levels of fairness on average, but be more *variable* as a group in how much they report to be fair. We tested this possibility, and a Fligner-Killeen test of homogeneity of variances suggested that variance in fairness standards did not differ between the veil of ignorance condition and the full knowledge condition, suggesting that participants' responses to the fairness standards question were a relatively accurate reflection of their personal standards in both conditions. We now report these results in the Supplemental Materials (pg. 5):

“We found that participants' beliefs about what constitutes a fair offer were no different when made under a veil of ignorance compared with full knowledge ($W = 6198, p = .20$), nor were the variances in these beliefs ($\chi^2_{\text{Fligner-Killeen}}(1) = .06, p = .81$). These findings suggest that our measure of personal standards was consistent across conditions, further supporting its construct validity.”

Regarding the possibility that some participants saw the question as a potential opportunity for strategic communication: since participants knew their responses were anonymous and they would have no chance to interact with their partner, we believe it is unlikely that participants were seeking to strategically communicate with anyone.

To further support the validity of our measure of personal standards, you suggested reporting the correlation between self-reported standards and generosity. We now report these correlations in the Methods sections of Experiments 2 & 3 on pg. 16 and 22:

“**Internal validity of personal standards measure.** To support the validity of our measure of personal standards of fairness, we tested how well this measure tracked with people's actual behavior. People's reported standard of fairness were highly correlated with their actual behavior ($r_s = .78, p < .001$)—suggesting that such standards reflected a meaningful guide for people's actual behavior (see Supplementary Text for additional analyses).”

“**Internal validity of personal standards measure.** To support the validity of our measure of personal standards of fairness, we again tested how well this measure tracked with people's actual behavior. In line with Experiment 2, people's reported standard of fairness correlates reasonably well with their actual behavior ($r_s = .55, p < .001$).”

R3.4: Can our effects be explained by a desire for internal consistency between what one says and does?

You questioned whether our observed results could be driven by a mere inconsistency between what one says and what one does—with the implication being that our effects might be observed regardless of what one’s standards are, or even in a non-moral domain.

There are at least three reasons that make this interpretation unlikely. First, models of cognitive dissonance that emphasize personal standards (Stone & Cooper, 2001) suggest that a motivation to misremember would be driven largely by inconsistency with a *personally valued* self-attribute, such as seeing oneself as moral (Mazar et al., 2008). However, we would not predict misremembering to occur in the face of behaving inconsistently with past statements in any domain. If someone violated a personally irrelevant statement they made (e.g., the number of times they stated versus actually did eat lunch at a specific restaurant), they would likely have little motivation to misremember their actions.

Second, while we made peoples’ fairness standards especially salient in Experiments 2 and 3 by asking participants about them directly, we did not explicitly ask participants about their fairness standards in Experiment 1. Nonetheless, we observed motivated misremembering specifically among selfish participants in Experiment 1. This suggests that, even when they do not state their standards, participants are likely aware of them, and violating those standards can lead to misremembering.

Finally, if misremembering is driven by a desire to maintain consistency between one’s stated standards and one’s behavior, we would expect to observe misremembering not only in participants who violated their fairness standards, but also in participants who exceeded their fairness standards. As mentioned in our response to Reviewer 2 above, we conducted new analyses that compared misremembering in *violators* (participants who gave less on average than what they said was fair) and *exceeders* (participants who gave more on average than what they said was fair). Crucially, one-sample signed rank tests showed no evidence of misremembering in exceeders (Experiment 2, $V = 538$, $p = .34$, $d = .03$, $\delta = -.07$; Experiment 3, $V = 1553$, $p = .80$, $d = .06$, $\delta = 0.02$). Instead, exceeders, unlike violators, appeared to be extremely accurate at recalling their decisions. These results are consistent with our motivated account, and inconsistent with the possibility that people misremember to maintain consistency with their stated personal standards. We report these new analyses in the Results section (p. 17; p. 22):

“we examined memory errors in exceeders ($N = 60$, average generosity = 30.3%)—a group that showed a similar degree of norm deviation and decision variance (See *SI Appendix*). In support of our hypothesis that misremembering is motivated, we found no evidence of misremembering towards the norm in exceeders ($V = 538$, $p = .34$, $d = .03$, $\delta = -.07$), and violators misremembered to a significantly greater degree exceeders ($W = 1209$, $p < .001$, $d = .58$, $\delta = .37$)”

R3.5: Clarifying participants' knowledge of the size of the endowment

You asked us to clarify whether participants knew the size of the endowment when deciding what percentage of it to allocate to the receiver or whether they simply allocated percentages of endowments whose size was unknown.

We have clarified this in the methods section of each experiment (See p. 8; p. 15; p. 21)

“As dictators, participants made decisions about an endowment that could range in size from CHF 10 to CHF 30, however its size was unknown at the time of choice... They were informed that one of their decisions would be randomly selected and implemented, and that the size of the endowment would also be randomly determined and revealed at the end of the experiment.”

“Moreover, participants made decisions about an endowment that could range in size from CHF 10 to CHF 30, however its size was unknown at the time of choice.”

“As dictators, participants made decisions about an endowment that could range in size from 10 to 30 cents. Like before, the size of this endowment was unknown at the time of choice.”

R3.6: Clarifying the format of choices in the experiments

You asked for more information about the graphical/numeric choice formats used in Experiments 1 and 2.

We have now added a supplemental figure that illustrates the choice formats used:

“Fig. S5. Choice format. Participants were asked to decide what percentage of the stake they would like to keep for themselves and what percentage of the stake they would like to send to the receiver. In the numeric format (A), each point represented 10% of the stake. In the graphical

format (B), each ball in the diagram represented 10% of the stake. The experiment was run in German; the figure depicts the English translation.”

R3.7: Clarifying the format of the recall measure

You asked us to clarify the format of the recall measure, including its visual format, increments, and whether the format of this scale/depiction was the same as the one in which subjects made the decisions in the first place.

We now include these details in the Method sections for each experiment (p. 9; p. 16; p. 22)

“The memory recall question was presented using the exact same format as the choice question. Thus, participants could only report their recalled average donation in 10% increments. Furthermore, the format of the recall question was identical to that in which they made the decisions in the first place.”

“As in Experiment 1, the memory recall question was presented using the exact same format as the choice question. Thus, participants could only report their recalled average donation in 10% increments. Furthermore, the format of the recall question was identical to that in which they made the decisions in the first place.”

“In Experiment 3, participants reported the percentage they recalled transferring on a sliding scale (the “Slider” Question type and “Bars” subtype in Qualtrics). They could report their recalled average in 1% increments on this scale. Furthermore, we used a slightly different scale/depiction from the format in which decisions were made. Decisions were made also using a similar but visually distinct slider scale (the “Slider” question type and “Sliders” subtype in Qualtrics), and were instead made in 10% increments.”

R3.8: Clarifying whether participants were aware of endowment size when recalling their decisions

You suggested that the recall instructions could be perceived as ambiguous if participants were aware of the endowment size at the time of recall, and asked whether it is possible that participants could be confused about whether the recall question probed the average amount given, or average percentage given.

Fortunately, since participants were unaware of the size of the endowment they were allocating at both the time of choice and at recall—participants did not have sufficient knowledge to engage in the alternative form of calculation you suggested (i.e., averaging over the monetary value of each decision rather than the percentage of the endowment). Furthermore, because in Experiments 1 and 2 the format of the recall question was identical to the format of the original decisions, this left no ambiguity in terms of how participants were expected to answer the recall question.

R3.9: Justifying our use of nonparametric statistics in Experiments 2 and 3

You asked whether there is independent justification for using nonparametric statistics in Experiments 2 and 3.

We apologize for the confusion. We have now clarified that for all three experiments, memory errors were non-normally distributed (see p. 10):

“Across all three experiments, memory errors were not normally distributed ($W_{shapiro-wilk} = .53-78, p < .001$), thus we report non-parametric statistics for all key comparisons.”

R3.10: Providing further details of recalled vs. actual generosity

You requested we report mean percentages of recalled vs actual generosity.

We have now added the mean percentages of recalled vs. actual giving into a table in the Supplementary Material (Table S1):

Table S1. Descriptive statistics for key comparisons across all three experiments.

	Actual giving		Recalled giving	
	M	SD	M	SD
Experiment 1 (N=109)				
1. All participants	26.68	18.74	29.24	20.76
2. Behaviorally selfish	10.11	9.55	13.76	16.14
3. Behaviorally generous	42.36	9.41	43.90	12.26
Experiment 2 (N=234)				
1. All participants	23.32	18.12	24.37	18.15
2. Violators	22.28	13.29	25.65	13.86
3. Upholders	30.84	15.17	30.85	15.08
Experiment 3 (N=604)				
1. All participants	31.21	20.71	32.01	20.88
2. Violators	15.97	16.65	18.72	19.15
3. Upholders	38.25	15.59	38.47	15.85

R3.11: Contextualizing findings with relevant literature and citing past work on accuracy

You asked for more detail on how the present research might be reconciled with past findings on the accuracy of moral violations. You also suggested that we highlight other work on memory accuracy, including studies that focus on accuracy of recalling moral rules (e.g., Shu et al’s work in which cheating participants were more likely to forget the content of a code of conduct they signed).

Thanks for this suggestion. We have added text to address both points in our Introduction (p. 5) and Discussion (p. 26):

“While there is evidence that people are less accurate at recounting relevant moral rules after cheating (Shu & Gino, 2012; Shu, Gino, & Bazerman, 2011), or relevant story details after hypothetical acts of cheating (Kouchaki & Gino, 2016; but see also: Stanley et al., 2018), it remains an open question whether people who truly violate their own standards misremember the details of their actions (Saucet & Villeval, 2018).”

“these findings contribute to a growing literature on the *motivated* nature of memory (Kouchaki & Gino, 2016; Rotella & Richeson, 2013; Saucet & Villeval, 2018, 2018; Shu & Gino, 2012; Shu et al., 2011; Stanley et al., 2018; Tappin & McKay, 2017; Tasimi & Johnson, 2015) by supporting the idea that people can misremember not just rules, or hypothetical situations, but also concrete actions.”...”Specifically, these findings suggest that those who violate (as opposed to uphold) their personal standards misremember the extent of their selfishness.”

R3.12: Fixing typos

You spotted a few typos in the manuscript.

These have been fixed. Thank you for catching these errors!

References

- Al-Aidroos, N., Said, C. P., & Turk-Browne, N. B. (2012). Top-down attention switches coupling between low-level and high-level areas of human visual cortex. *Proceedings of the National Academy of Sciences, 109*(36), 14675–14680.
<https://doi.org/10.1073/pnas.1202095109>
- Amodio, D. M. (2019). Social Cognition 2.0: An Interactive Memory Systems Account. *Trends in Cognitive Sciences, 23*(1), 21–33.
<https://doi.org/10.1016/j.tics.2018.10.002>
- Anderson, M. C., & Green, C. (2001). Suppressing unwanted memories by executive control. *Nature, 410*(6826), 366.
- Anderson, M. C., & Hanslmayr, S. (2014). Neural mechanisms of motivated forgetting. *Trends in Cognitive Sciences, 18*(6), 279–292.
- Bem, D. J. (1972). Self-perception theory. In *Advances in experimental social psychology* (Vol. 6, pp. 1–62). Elsevier.
- Bénabou, R., & Tirole, J. (2011). Identity, morals, and taboos: Beliefs as assets. *The Quarterly Journal of Economics, 126*(2), 805–855.
- Benoit, R. G., & Anderson, M. C. (2012). Opposing mechanisms support the voluntary forgetting of unwanted memories. *Neuron, 76*(2), 450–460.
- Chambers, J. R., & Windschitl, P. D. (2004). Biases in social comparative judgments: the role of nonmotivated factors in above-average and comparative-optimism effects. *Psychological Bulletin, 130*(5), 813.

- Chun, M. M., & Turk-Browne, N. B. (2007). Interactions between attention and memory. *Current Opinion in Neurobiology*, *17*(2), 177–184.
<https://doi.org/10.1016/j.conb.2007.03.005>
- Crowder, R. G. (2014). *Principles of learning and memory: Classic edition*. Psychology Press.
- Dana, J., Weber, R. A., & Kuang, J. X. (2007). Exploiting moral wiggle room: experiments demonstrating an illusory preference for fairness. *Economic Theory*, *33*(1), 67–80.
- Exley, C. L. (2015). Excusing selfishness in charitable giving: The role of risk. *The Review of Economic Studies*, *83*(2), 587–628.
- Fehr, E., & Rangel, A. (2011). Neuroeconomic Foundations of Economic Choice—Recent Advances. *Journal of Economic Perspectives*, *25*(4), 3–30.
- Feiler, L. (2014). Testing models of information avoidance with binary choice dictator games. *Journal of Economic Psychology*, *45*, 253–267.
- Fiske, S. T., & Taylor, S. E. (1984). *Social Cognition*. Random House.
- Gagnepain, P., Henson, R. N., & Anderson, M. C. (2014). Suppressing unwanted memories reduces their unconscious influence via targeted cortical inhibition. *Proceedings of the National Academy of Sciences*, *111*(13), E1310–E1319.
- Gneezy, U., Imas, A., & Madarász, K. (2014). Conscience accounting: Emotion dynamics and social behavior. *Management Science*, *60*(11), 2645–2658.
- Gordon, R., Franklin, N., & Beck, J. (2005). Wishful thinking and source monitoring. *Memory & Cognition*, *33*(3), 418–429.

- Gruber, M. J., Ritchey, M., Wang, S.-F., Doss, M. K., & Ranganath, C. (2016). Post-learning Hippocampal Dynamics Promote Preferential Retention of Rewarding Events. *Neuron*, *89*(5), 1110–1120. <https://doi.org/10.1016/j.neuron.2016.01.017>
- Hu, X., Bergström, Z. M., Bodenhausen, G. V., & Rosenfeld, J. P. (2015). Suppressing Unwanted Autobiographical Memories Reduces Their Automatic Influences: Evidence From Electrophysiology and an Implicit Autobiographical Memory Test. *Psychological Science*, *26*(7), 1098–1106. <https://doi.org/10.1177/0956797615575734>
- Johnson, M. K. (2006). Memory and reality. *American Psychologist*, *61*(8), 760.
- Kappes, A., & Crockett, M. J. (2016). The benefits and costs of a rose-colored hindsight. *Trends in Cognitive Sciences*, *20*(9), 644–646.
- Kouchaki, M., & Gino, F. (2016). Memories of unethical actions become obfuscated over time. *Proceedings of the National Academy of Sciences*, *113*(22), 6166–6171.
- Kruger, J. (1999). Lake Wobegon be gone! The "below-average effect" and the egocentric nature of comparative ability judgments. *Journal of Personality and Social Psychology*, *77*(2), 221.
- Kunda, Z. (1990). The case for motivated reasoning. *Psychological Bulletin*, *108*(3), 480.
- Larson, T., & Capra, C. M. (2009). Exploiting moral wiggle room: Illusory preference for fairness? A comment. *Judgment and Decision Making*, *4*(6), 467.
- Malle, B. F. (2006). The actor-observer asymmetry in attribution: A (surprising) meta-analysis. *Psychological Bulletin*, *132*(6), 895.

- Manning, J. R., Hulbert, J. C., Williams, J., Piloto, L., Sahakyan, L., & Norman, K. A. (2016). A neural signature of contextually mediated intentional forgetting. *Psychonomic Bulletin & Review*, *23*(5), 1534–1542.
- Mazar, N., Amir, O., & Ariely, D. (2008). The dishonesty of honest people: A theory of self-concept maintenance. *Journal of Marketing Research*, *45*(6), 633–644.
- Moutoussis, M., Dolan, R. J., & Dayan, P. (2016). How People Use Social Information to Find out What to Want in the Paradigmatic Case of Inter-temporal Preferences. *PLOS Computational Biology*, *12*(7), e1004965.
<https://doi.org/10.1371/journal.pcbi.1004965>
- Mullen, E., & Monin, B. (2016). Consistency versus licensing effects of past moral behavior. *Annual Review of Psychology*, *67*.
- Ross, M., & Wilson, A. E. (2003). Autobiographical memory and conceptions of self: Getting better all the time. *Current Directions in Psychological Science*, *12*(2), 66–69.
- Rotella, K. N., & Richeson, J. A. (2013). Motivated to “forget” the effects of in-group wrongdoing on memory and collective guilt. *Social Psychological and Personality Science*, *4*(6), 730–737.
- Saucet, C., & Villeval, M. C. (2018). Motivated Memory in Dictator Games.
<https://doi.org/10.1177/0956797618781335>
- Shalvi, S., Gino, F., Barkan, R., & Ayal, S. (2015). Self-serving justifications: Doing wrong and feeling moral. *Current Directions in Psychological Science*, *24*(2), 125–130.

- Shu, L. L., & Gino, F. (2012). Sweeping dishonesty under the rug: How unethical actions lead to forgetting of moral rules. *Journal of Personality and Social Psychology*, *102*(6), 1164.
- Shu, L. L., Gino, F., & Bazerman, M. H. (2011). Dishonest deed, clear conscience: When cheating leads to moral disengagement and motivated forgetting. *Personality and Social Psychology Bulletin*, *37*(3), 330–349.
- Stanley, M. L., Henne, P., Iyengar, V., Sinnott-Armstrong, W., & De Brigard, F. (2017). I'm not the person I used to be: The self and autobiographical memories of immoral actions. *Journal of Experimental Psychology: General*, *146*(6), 884.
- Stanley, M. L., Yang, B. W., & Brigard, F. D. (2018). No evidence for unethical amnesia for imagined actions: A failed replication and extension. *Memory & Cognition*, 1–9. <https://doi.org/10.3758/s13421-018-0803-y>
- Tappin, B. M., & McKay, R. T. (2017). The illusion of moral superiority. *Social Psychological and Personality Science*, *8*(6), 623–631.
- Tasimi, A., & Johnson, M. K. (2015). A self-serving bias in children's memories? *Journal of Experimental Psychology: General*, *144*(3), 528.
- Tversky, A., & Kahneman, D. (1981). The framing of decisions and the psychology of choice. *Science*, *211*(4481), 453–458.
- Walster, E., Berscheid, E., & Walster, G. W. (1973). New directions in equity research. *Journal of Personality and Social Psychology*, *25*(2), 151.

Reviewers' comments:

Reviewer #1 (Remarks to the Author):

Good job responding to the reviews. I am satisfied. Congratulations on a nice paper.

-Roy F. Baumeister

Reviewer #2 (Remarks to the Author):

The authors clearly put a ton of work into their revision and response letter for this paper, and I've spent a lot of time in turn trying to sift through it. I've appreciated all of the work the authors put into this, and have enjoyed thinking hard about their paper. In the interests of saving a little time, given how much of it we've all spent so far on this, I will try to be as concise as I can.

In my evaluation of the authors' first submission, I raised an obvious alternative interpretation based on cognitive consistency. I think the authors have addressed this well. This mechanism does not seem to be operating in any obvious fashion, and I think the additional scholarship in the paper now makes it stronger.

However, finding the absence of evidence for one mechanism does not automatically provide evidence for another, as the authors imply in several places. On page 24, for instance, the authors write that failing to find evidence for norm-consistent memory "support[s] the idea that misremembering is indeed motivated." Not so. Negative evidence is not positive evidence for an orthogonal mechanism. Motivated forgetting makes no prediction about this.

The lack of any direct positive evidence for a clearly articulated mechanism, in the end, continues to be one of my primary concerns. I mentioned in my last review that the paper would be much stronger if the authors actually articulated a clear goal-directed (i.e., motivated) mechanism for their evidence and then provided a direct test of it, with positive evidence rather than what I think amounts to largely circumstantial evidence in this paper. In fact, the authors' attempt to clarify their mechanism by distinguishing between motivated reasoning and motivated memory only muddies matters further. If motivated forgetting is not guided by the same kinds of processes known to guide other types of motivated reasoning, then what is guiding it? What is the actual process? The authors cite suppression in the introduction, which would itself be undergirded by some sort of motivated reasoning process. And then in their response letter, the authors explicitly state that they are using motivated reasoning to explain motivated forgetting: "That is, we suggest that people's desire to reduce dissonance can shape memories of concrete actions." Huh? What? I am befuddled. Cognitive dissonance IS a mechanism for motivated forgetting, as tested in the cited Kouchaki & Gino (2016) paper. This mechanism specifies a clear process whereby people feel discomfort behaving worse than their personal standard that then leads to a misremembering of prior behavior and subsequent reduction in negative affect. The authors instead suggest that motivated forgetting would preclude motivated reasoning, but then their later

writing renders this nonsense because they claim that motivated reasoning is a goal-directed attempt to reduce negative feeling (Experiment 3). A goal-directed, dissonance-reducing, model is never directly tested in this paper, never clearly proposed, and somehow seemingly excluded from the hair splitting done in the introduction to distinguish between motivated reasoning and motivated forgetting.

More troubling for me is that the papers cited in support of motivated forgetting in the introduction are cited in a misleading way, and the evidence in these papers do not actually seem like they would predict the results in the present paper. In particular, most (possibly all, but I can't claim to have read it all!) research in the motivated forgetting literature explicitly instructs people to suppress a memory, rather than finding that suppression of an unwanted event arises spontaneously when people encounter an unwanted memory. I believe this is true of the studies I saw reviewed by Anderson & Hanslmayer, but it is definitely true in the Hu et al (2015) paper cited at the same point on page 5 of the current manuscript. The experiment in Hu et al. explicitly instructs people to suppress an unwanted memory, and they find no evidence of such suppression in the absence of the explicit instruction. The cited evidence does not suggest, on my read of the work, that suppression would arise spontaneously among those who have behaved immorally as they seem to be implying will happen in the experiments reported in this paper. In addition, the paper cited next on page 5 (Kouchaki & Gino, 2015) as showing motivated misremembering actually tests in a rather compelling way a dissonance-based mechanism of motivated forgetting, yet the authors do not cite it correctly. Indeed, this paper suggests that you would NOT find motivated forgetting among the participants in these experiments. Kouchaki & Gino find evidence of motivated forgetting ONLY after a delay following unethical behavior (or consideration of unethical behavior) ranging from a few days to a couple of weeks. In one experiment (#5) of that paper, they find no evidence for motivated forgetting following a distractor task and a 30-minute delay. They only find significant evidence following a 4-day delay. There is no delay period in the current experiments, which this prior research suggests is necessary if the process the current authors are citing this paper for is at work. Perhaps the authors did not read these papers carefully enough to notice that they do not actually support the predictions made in this paper? Seeing them cited in a way that seemed misleading to me was quite troubling.

Prior research has found evidence for motivated forgetting of unethical behavior, including a careful articulation of an underlying motivated reasoning mechanism for it and a direct test of it (e.g., Kouchaki & Gino, 2015). The authors' current manuscript obfuscates the precise mechanism and tries to draw an artificial distinction between processes of motivated reasoning and processes of motivated forgetting. The current demonstrations do provide an advance on this prior work by identifying a systematic misremembering of prior behavior, rather than just forgetting, but that contribution alone is modest and incremental rather than really substantial. In the end, though, their key independent variable is nonexperimental, and hence causal claims should be made only very tentatively. These experiments do not experimentally manipulate whether a person's behavior falls show of his or her personal standards but rather simply measures it. Those who fall short of their own stated standards are likely to differ in lots of ways from those who don't. Maybe violators just have worse memories in general, which is consistent with their tendency to make larger errors in at least two of the reported experiments. Maybe their ethical standards are just weaker to begin with, or stronger to begin with, or any number of other

possible interpretations we could raise. By not experimentally manipulating the critical independent variable, specifying the dissonance-reducing process they are going through to misremember a prior behavior, and then measuring (or manipulating) it in an actual experiment, we simply can't know why the effects are arising in this paper. And simply playing whack-a-mole to rule out one alternative without providing positive causal evidence in favor of a mechanism simply does not provide a strong empirical contribution.

Some of the reported evidence in the paper is certainly consistent with a motivated reasoning account, but not as consistent as the authors explain. For instance, I liked the mood measure in Experiment 3 very much, but the authors overstate its importance. They write, "These results favor a motivated account over a rational inference account by virtue of their specificity: they support the idea that misremembering is serving the goal of reducing psychological discomfort associated with a threatened moral self-image." No, they do not, because they never measured psychological discomfort at an earlier part of the experiment. They only find that those who correctly remember violating their own ethical standards feel worse at the end of the experiment than everyone else. In addition, wouldn't the authors' motivated forgetting mechanism at least require some time to unfold, thereby leading decisions among violators to take longer than everyone else?

I could go on, but I actually think this matter is fairly straightforward. For these experiments to provide a contribution significant enough to be published in this journal, I think they need to specify their mechanism in clear (rather than confusing) detail, manipulate their key IV experimentally, and then measure (or manipulate) the proposed mechanism. Without this, we do not actually know that violating one's ethical standards lead people to misremember their behavior in order to maintain a positive self-image, which is the paper's most fundamental claim.

Reviewer #3 (Remarks to the Author):

In this revision, the authors have made several efforts to address the considerations raised in the last round of review, including providing helpful clarifications and new analyses. Ultimately, I think the authors have done what they can do given their design and the data they have. At the same time, one of three main concerns is still present, albeit diminished somewhat in magnitude.

1 . Motivated mis-remembering vs difficulty of remembering (or difficulty of averaging)

One primary concern was that reduced memory for the choices of norm violators could have been due to the lower memorability of the sets of decisions by those participants (relative to non-violator participants). (For example, the former decision sets might have been more variable, making them more difficult to remember and/or more difficult to average in one's head.) The authors took several sensible steps to try to address this alternative explanation, which do help but do not rule it out.

a. One approach to ruling out the structure of the decision sets as an alternative explanation was to re-analyze the data while excluding responses from “static deciders”, i.e., participants who made the same choice on every trial. Although the authors report in the response letter that “all key results remain the same in these analyses”, the results table does not seem to be fully consistent with this claim. For example, in Experiment 1, when all participants are included, there is a significant difference in memory errors by selfish vs generous participants but no significant difference in memory accuracy. In the new analysis in which static deciders are excluded, the reverse is reported: there is no significant difference in memory errors but a significant difference in memory accuracy. Is this correct? If so, should this be interpreted as the key results remaining the same? The simplest explanation is that the rows are mislabeled in the new table, but one way or another this should presumably be addressed.

b. Minor point: my understanding is that there were no “violators” or “upholders” in Experiment 1, just selfish vs generous participants – but the new table labels the groups as violators and upholders.

c. The authors took a clever, second approach to ruling out decision variance as an alternative explanation: comparing norm violators to norm exceeders. Despite both deviating from their stated norms, these groups showed different levels of memory (with forgetting occurring more in the violators). This also helps somewhat, but unfortunately violators’ decisions still varied to a greater degree than exceeders, meaning that this analysis can’t fully address the concern either.

Overall, I think the authors have done what they can on this point given the existing data, but there still seems to be room for the possibility that the sets of the decisions were just more difficult to remember or to average together. I continue to be hesitant to suggest an additional study that could rule this out more directly (e.g., as described in my review of the original manuscript), but at the very least I do suggest that this be noted as an alternative that can’t be fully ruled out under the circumstances, which the authors have added to the discussion.

Reviewer Replies

Reviewer #1

R1.1: You wrote "Good job responding to the reviews. I am satisfied. Congratulations on a nice paper.

- Roy F. Baumeister"

Response:

Thank you for your encouraging feedback and helpful review, Roy!

Reviewer #2

R2.1: You agree that we have ruled out the alternative interpretation based on cognitive consistency. Specifically, you wrote “I think the authors have addressed this well. This mechanism does not seem to be operating in any obvious fashion, and I think the additional scholarship in the paper now makes it stronger”.

Response:

We thank the reviewer for their positive remarks, and we are pleased to hear that they think the alternative cognitive account was well addressed in our revision.

R2.2: You note that ruling out the cognitive consistency mechanism does not provide positive evidence for a motivated mechanism.

Response:

We thank the reviewer for pointing out that our language is sometimes inferentially too strong. We have revised main text to modify the occurrences of such language, using the phrase “consistent with” instead of “supports”. With respect to the specific quote you mentioned (pg. 24), it now reads as follows:

“we found no evidence of a norm-directed memory errors in exceeders...and violators misremembered to a significantly greater degree than exceeders...which **is consistent with** the idea that misremembering is motivated”

We also fully agree that ruling out the cognitive consistency mechanism does not provide positive evidence for a motivated mechanism. We have now run additional experiments to provide this positive evidence—see response to Comment R2.5 below for further details.

R2.3: You asked us to specify a clear mechanism for motivated misremembering. In particular, you wrote “If motivated forgetting is not guided by the same kinds of processes known to guide other types of motivated reasoning, then what is guiding it? What is the actual process? The authors cite suppression in the introduction, which would itself be undergirded by some sort of motivated reasoning process”.

Response:

On this point, the reviewer’s comments indicate an important misunderstanding: our sense is that we are using the term “motivated reasoning” in a different way than the reviewer, which we believe has resulted in them misinterpreting key claims in our paper.

Our usage of the term motivated reasoning. We believe it is uncontroversial that episodic *memory* phenomena (e.g., encoding, storage, and retrieval of concrete events) are dissociable from *reasoning* phenomena (e.g., thinking logically about abstract, semantic content, such as attitudes or beliefs; Tulving, 1972). We also believe that

motives can influence both sets of phenomena based on prior work (Kunda, 1990; Anderson & Hanslmayr, 2014). Additionally, we maintain that each of these phenomena would manifest in distinct ways in our experiment. Motivated reasoning would involve accurate recall, but a self-serving *justification* of one's actions (e.g., a person might believe that transferring only 40% instead of 50% is fair because they need the money more than their partner). On the other hand, motivated misremembering would involve keeping one's norm intact, but *distorting* one's memory of their behavior to match their norm (e.g., recalling transferring 50%, instead of the true 40%).

We apologize that it was unclear in our work how these mechanisms are distinct. We hope this is clarified above, and in the text below (pg. 4; changes in blue):

“Social scientists have often credited our ability to engage in *motivated reasoning* (Kunda, 1990)—that is, we form self-serving beliefs and attitudes to justify immoral acts to ourselves before or after the events unfold (Mazar, Amir, & Ariely, 2008; Shalvi, Gino, Barkan, & Ayal, 2015; Walster, Berscheid, & Walster, 1973). This feat is accomplished in a number of ways. For one, people tend to strike a justifiable balance between self-interest and their moral values – for instance, lying just enough to profit financially, but not so much as to harm their moral self-image (Mazar et al., 2008). In addition, people psychologically *distance* themselves from their unethical actions – attributing past misdeeds to situational pressures (Malle, 2006), or having been a “different person” at the time (Ross & Wilson, 2003; Stanley, Henne, Iyengar, Sinnott-Armstrong, & De Brigard, 2017). Moreover, people exploit *uncertainty* – behaving more selfishly when the consequences for others are ambiguous (Exley, 2015), and avoiding information about how their actions may have harmed others (Dana, Weber, & Kuang, 2007; Feiler, 2014; Larson & Capra, 2009). A common thread in each of these self-serving strategies is that they operate over abstract beliefs and attitudes.

Another possibility that has received less attention is that our desire to believe we are moral may distort memories of our concrete experiences. When people's actions fall short of their personal standards, they might misremember acting in line with those standards. Misremembering past immoral behavior as moral would *pre-empt* the need to engage in motivated reasoning, because behavior that falls short of one's personal standards would instead be revised in memory.”

Our interpretation of the reviewer's usage of the term motivated reasoning. It now seems clear to us that the reviewer has been using the term ‘motivated reasoning’ not to refer to the psychological phenomena above, but to a *motivation* (and namely, *dissonance motivation*). This is evident from the reviewer's following statements above, and the comment below:

“in their response letter, the authors explicitly state that they are using motivated reasoning to explain motivated forgetting: “That is, we suggest that people's desire to reduce dissonance can shape memories of concrete actions. “ Huh? What? I am befuddled.”

The reviewer's statement does not follow from our interpretation of motivated reasoning (Kunda, 1990). Possessing a desire to reduce dissonance (e.g., wanting to reduce

discomfort felt about an unfair act) does not require, and is certainly not equivalent to, motivated reasoning (e.g., rationalizing one's unfair act as fair). Our findings explicitly challenge the former assumption—which is part of why we believe this work will have an important impact on psychological theory.

However, if the phrase “motivated reasoning” in the reviewer’s comments is treated as equivalent to “motivation to reduce dissonance”, the reviewer’s above statement (and much of their current and prior review) are much clearer. As such, we have addressed this concern by providing clear and direct evidence that this motivation is active (see response to Comment 2.5).

R2.4: You asked us to clarify differences between past work on motivated forgetting and the current work.

Response:

We thank the reviewer for highlighting another important point of confusion. We apologize for the misunderstanding. To be clear, it was not our intention to be misleading. We cite Kouchaki & Gino (2015) as well as Hu et al (2015) only to demonstrate that motivated memory phenomena are *possible*. There are of course many potential mechanisms by which motivated memory can operate (Anderson & Hanslmayr, 2014). Our aim was not to replicate the exact phenomenon demonstrated by Kouchaki and Gino’s work, but to demonstrate a novel one. In our previous manuscript, we already distinguished the phenomenon in our work from Kouchaki and Gino’s with the paragraph below (pg. 5, relevant text in bold):

“Importantly, accuracy and vividness of memories are dissociable: impaired vividness (e.g., foggy mental imagery for the moment you tipped a barista yesterday) does not necessarily imply impaired accuracy (e.g., misremembering how much you actually tipped the barista), and vice-versa (20, 31, 32). Vividness and accuracy are not only psychologically dissociable, but also neurally dissociable at memory retrieval (33). **This leaves open the question of whether memory accuracy may serve as another mechanism through which individuals can act selfishly and ultimately still feel moral.** When behaving unfairly (e.g., giving a stingy tip), people may misremember the extent of their selfishness, thus preserving the view that they treat others fairly. To our knowledge, it is unknown whether people engage in such motivated misremembering of selfish actions – that is, remembering having been more generous than they truly were”.

We believe that the fact that these authors’ observations differ from the one we show is in fact a *strength* of our work. We also believe the reviewer, at least in part, agrees about the novelty of our mechanism, as they wrote: “*The current demonstrations do provide an advance on this prior work by identifying a systematic misremembering of prior behavior, rather than just forgetting...*”.

R2.5: You asked us to experimentally manipulate the motivation to misremember and demonstrate a causal effect on misremembering.

Response:

We agree that a major limitation of our original experiments was that they did not provide an experimental manipulation of the motivation to misremember. We also agree that the extent to which the mood measure in Experiment 3 supports a motivated account was overstated, and in our revision, we have therefore de-emphasized a reliance on mood measures to support a motivated account. Instead, as you suggested, we introduced a stronger test with new Experiments 4a and 4b (total N = 2191), which experimentally manipulated the motivation to misremember. Our experimental manipulation of the motivation to misremember provides conclusive support for a motivated account: we observe motivated misremembering among people who freely chose to violate their norm, but not among those who were forced to violate their norm and did not feel responsible for doing so. For clarity, in the revised manuscript we have moved all of the mood measures to the supplementary materials for interested readers, and focus on our experimental manipulation of motivation in the main text. Below, we provide text that offers further detail on Experiments 4a and 4b:

Experiment 3 showed that even when incentivized with a monetary bonus that directly scaled with memory accuracy, violators still systematically make self-serving memory errors. These results are consistent with the possibility of motivated misremembering. However, they only demonstrate this phenomenon indirectly. Indeed, a motivated account of misremembering would receive compelling evidence over a rational inference account with direct evidence for two key conditions.

First, motivated misremembering should be *motive-dependent*. That is, misremembering should occur only when people have a motive to misremember, and should cease to occur if motives to misremember are removed. In particular, a rich, long standing literature on motivated cognition suggests that when people no longer view themselves as *personally responsible* for a dissonance-inducing action, they no longer experience dissonance motivation (Cooper & Fazio, 1984; Gosling, Denizeau, & Oberlé, 2006; Scher & Cooper, 1989; Wicklund & Brehm, 1976). Indeed, feelings of responsibility for an action serve as a bridge between the action and one's self-concept (Thibodeau & Aronson, 1992). As such, only if people feel personally responsible for their fairness violation should they experience a moral self-threat, and become motivated to reduce it via misremembering.

Second, motivated misremembering should be largely *choice-independent*. One possible alternative explanation for our findings is that there is something about the pattern of violators' realized choices that makes those patterns more prone to being misremembered—for instance, containing more variable transfer amounts. Thus, it is important to rule out the possibility that the sets of choices made by violators are just inherently more prone to being misremembered than the sets of choices made by upholders.

In a final pair of experiments (Experiments 4a and 4b), we examined these two hypotheses to directly test the motivated nature of misremembering. In Experiment 4a, an initial wave of participants (*free-choice* deciders) freely decided how to split money with anonymous partners, and later recalled their choices. In Experiment 4b, a second

wave of participants (*forced-choice* deciders) were each yoked to a randomly selected free-choice decider from Experiment 4a, and were forced to make (and later recall) the exact same set of transfer choices as their yoked free-choice decider (See Figure S6). Crucially, in Experiment 4b, forced-choice deciders (and receivers) were instructed that the decider had *no responsibility* for their transfer choices—thereby removing any incentive to misremember their choices. Moreover, we confirmed that feelings of personal responsibility were lower among forced-choice (versus free-choice) deciders.

We made the following key predictions. Based on the motive-dependence hypothesis, we predicted that those who freely violated their own standards would make self-serving memory errors, whereas those who were forced to violate their own standards and perceived no personal responsibility for their actions would show no such bias toward self-serving memory errors.

Similarly, based on the choice-independence hypothesis, we predicted that participants asked to recall the choice sets freely made by violators would be no more likely to make self-serving memory errors than participants asked to recall the choice sets freely made by upholders.

We indeed confirmed both the motive-dependence and choice-independence predictions. Forced-choice violators who perceived no responsibility for their actions showed no self-serving bias in memory, whereas free choice violators did. Moreover, violators' choice sets were no more prone to being misremembered than upholders when recalled by a naïve observer (i.e. a forced-choice participant). See results below (pgs. 32-36):

Motive-dependence. To confirm the efficacy of our manipulation, we first assessed how *responsible* participants perceived themselves to be for their actions under free-choice (Experiment 4a) and forced-choice (Experiment 4b) conditions. As predicted, free-choice participants reported a high degree of responsibility for their actions ($M = 6.28$, $SD = 1.18$), with 64% ($N = 452/709$) reporting the highest level of responsibility possible for their actions (7 = “Extremely responsible”; see Figure S6A). In contrast, forced-choice participants reported a low degree of responsibility for their actions ($M = 2.14$, $SD = 2.03$), with 71% ($N = 409/580$) reporting the lowest level of responsibility possible for their actions (1 = “Not at all responsible”; see Figure S6B). The difference in ratings between these two groups was significant ($W = 24374223.5$, $p < .001$, $d = 2.57$, $\delta = 83$).

We predicted that while those who freely violated their own standards would make self-serving memory errors, those who were forced to violate their standards (and consequently perceived no personal responsibility for their actions) should show no such bias toward self-serving memory errors.

Consistent with our predictions (and our prior experiments), in Experiment 4a, free-choice participants showed a systematic bias towards self-serving memory errors ($V = 29581$, $p < .001$, $d = .18$, $\delta = .08$; *SI Appendix*, Table S1). In particular, violators ($N = 231$, average generosity = 28.9%) recalled being more generous than they actually were ($V = 8609.5$, $p < .001$, $d = .40$, $\delta = .26$; Fig. 1D), whereas upholders ($N = 478$, average generosity = 40.5%) showed no such bias ($V = 6102.5$, $p = .92$, $d = .02$, $\delta = -.01$). We also found that violators showed a greater bias toward self-serving memory errors than upholders ($W = 42117$, $p < .001$, $d = .43$, $\delta = .24$), and were less accurate than

upholders—making significantly larger memory errors ($W = 37086$, $p < .001$, $d = .41$, $\delta = .33$; *SI Appendix*, Fig. S3A [in progress]).

In Experiment 4b, despite our instructions that sought to minimize feelings of responsibility in the forced choice experiment, we nevertheless observed some heterogeneity in reported responsibility, indicating that some participants felt responsible for their choices despite being forced to make them. Crucially, the motive dependence hypothesis predicts that the forced choice manipulation should eliminate motivated misremembering only in those violators who viewed themselves as *not personally responsible* for their actions, whereas those who viewed themselves as personally responsible should remain motivated toward a self-serving bias. To test this possibility, we independently assessed the memories of those who reported being “not at all responsible” for their actions (i.e., 1 out of 7 on our personal responsibility measure; $N = 409$), and those who self-reported some degree of personal responsibility for their actions (i.e., greater than 1 out of 7 on our personal responsibility measure; $M = 4.87$, $SD = 1.84$; $N = 171$).

Consistent with the motive-dependence hypothesis, forced-choice deciders who nonetheless perceived themselves as responsible for their actions showed a significant bias toward self-serving memory errors ($V = 3206.5$, $p = .019$, $d = .20$, $\delta = .14$; *SI Appendix*, Table S1). More specifically, violators ($N = 138$, average generosity = 20.0%) showed a significant bias toward self-serving memory errors ($V = 1258.5$, $p < .001$, $d = .50$, $\delta = .46$; Fig. 3B), whereas upholders ($N = 271$, average generosity = 45.6%) showed no such bias ($V = 372$, $p = .066$, $d = .08$, $\delta = -.13$). We also found that violators showed a greater bias toward self-serving memory errors than upholders ($W = 2010$, $p < .001$, $d = .61$, $\delta = .45$), and were less accurate than upholders – making significantly larger memory errors ($W = 2778$, $p = .006$, $d = .32$, $\delta = .24$; *SI Appendix*, Fig. S3B [in progress]).

Fig. 3. Motive-dependence of self-serving memory errors. (A) In Experiment 4b, forced-choice violators who perceived no responsibility for their actions showed no self-serving memory errors. (B) In contrast, forced-choice violators who nonetheless still perceived themselves as responsible for their actions exhibited self-serving memory errors. Error bars represent standard error of the mean (SEM). This figure is re-visualized with all raw data points in the *SI Appendix* (Fig. S3 [in progress]).

In contrast, forced-choice participants who reported feeling *no responsibility* for their actions showed no bias toward self-serving memory errors ($V = 4907.5$, $p = .73$, $d = .03$, $\delta = .01$; *SI Appendix*, Table S1). In particular, neither violators ($N = 138$, average generosity = 20.0%; $V = 874$, $p = .61$, $d = .05$, $\delta = -.02$; Fig. 3A) nor upholders ($N = 271$,

average generosity = 45.6%; $V = 372$, $p = .066$, $d = .08$, $\delta = -.13$) showed any bias toward self-serving memory errors. Moreover, violators and upholders did not differ in the extent of their self-serving memory errors ($W = 19297.5$, $p = .53$, $d = .01$, $\delta = .03$)—however violators made significantly larger memory errors ($W = 16123$, $p = .007$, $d = .01$, $\delta = .14$; *SI Appendix*, Fig. S3A [in progress]).

Crucially, responsible violators reported significantly larger self-serving memory errors than those that perceived no responsibility ($W = 7510$, $p < .001$, $d = .75$, $\delta = .40$). Moreover, violators in the free choice study made significantly larger self-serving memory errors than no-responsibility violators in the forced choice study ($W = 65402$, $p < .001$, $d = .23$, $\delta = .17$).

Choice-independence. We also predicted that naïve observers asked to recall the choice sets freely made by violators would be no more likely to make self-serving memory errors than naïve observers asked to recall the choice sets freely made by upholders. That is, while those who freely violate their own standards should tend to make self-serving memory errors, those forced to make the *same choices* would show *no bias* toward self-serving memory errors—regardless of whether those choices violated their own standards or not. To test this, we directly compared the memories of 580 matched pairs of free-choice deciders and forced-choice deciders.

When we examined the subset of free-choice deciders that were yoked with Experiment 4b forced-choice deciders ($N = 580$), we again observed a systematic bias towards self-serving memory errors ($V = 20118$, $p < .001$, $d = 0.19$, $\delta = 0.09$; *SI Appendix*, Table S1 [in progress]). In particular, violators ($N = 181$, average generosity = 29.0%) recalled being more generous than they actually were ($V = 5627.5$, $p < .001$, $d = 0.42$, $\delta = 0.26$; Fig. 4A), whereas upholders ($N = 398$, average generosity = 40.7%) showed no such bias ($V = 4255.5$, $p = 0.53$, $d = 0.04$, $\delta = 0.01$). We also found that violators showed a greater bias toward self-serving memory errors than upholders ($W = 27721.5$, $p < .001$, $d = 0.44$, $\delta = 0.23$), and were less accurate than upholders—making significantly larger memory errors ($W = 23396.5$, $p < .001$, $d = 0.43$, $\delta = 0.35$; *SI Appendix*, Fig. S3A [in progress]).

Fig. 4. Choice-independence of self-serving memory errors. (A) In Experiment 4a, free-choice violators tended to exhibit self-serving memory errors. (B) In contrast, in Experiment 4b, participants forced to make identical choices as violators exhibited no such bias toward self-serving memory errors. Error bars represent standard error of the mean (SEM). This figure is re-visualized with all raw data points in the *SI Appendix* (Fig. S4 [in progress]).

In contrast, yoked forced-choice deciders – who were forced to make, and then remember, the exact same choice sets as participants in the free-choice group -- showed no significant memory bias ($V = 16263$, $p = 0.15$, $d = 0.11$, $\delta = 0.05$; *SI Appendix*, Table S1 [in progress]). Indeed, neither those forced to make and then recall the same choices as violators ($N = 181$, average generosity = 29.0%; $V = 1608$, $p = 0.59$, $d = 0.08$, $\delta = 0.07$) nor upholders ($N = 398$, average generosity = 40.7%; $V = 7581.5$, $p = 0.18$, $d = 0.12$, $\delta = 0.04$) showed self-serving memory errors (Fig. 4B). While those paired with violators tended to make larger memory errors in general ($W = 23396.5$, $p < .001$, $d = 0.43$, $\delta = 0.35$; *SI Appendix*, Fig. S3A [in progress]), we found *no difference* in the extent to which yoked forced-choice deciders paired with violators (versus upholders) made self-serving memory errors ($W = 35482.5$, $p = 0.75$, $d = 0.06$, $\delta = 0.01$).

As mentioned, we believe these findings provide clear and strong evidence that misremembering observed in our experiments is in fact motivated.

R2.6: You asked whether motivated forgetting would require time to unfold.

Response:

As mentioned above, there are numerous mechanisms by which memory operates. Also, if a time delay were necessary, we would not observe misremembering effects in our study, which we do. Lastly, numerous studies suggest people can make short-term memory errors (MacLeod & Macrae, 2001; Macrae & MacLeod, 1999; Van den Berg, Shin, Chou, George, & Ma, 2012).

Reviewer #3

R3.1: You wrote that our new analyses and clarifications were helpful.

Response:

We thank the reviewer for their encouraging remarks.

R3.2: You expressed concern that two results in Experiment 1 were not consistent when “static deciders” were excluded. Specifically, you wrote: “In the new analysis in which static deciders are excluded...there is no significant difference in memory errors but a significant difference in memory accuracy”

Response:

We believe there is a misunderstanding. In Experiment 1, the original analyses and new analyses do show consistent findings. As can be seen in Table S6, there are no significant differences for either comparison between selfish vs generous participants in Experiment 1. Thus, our key results do remain the same in Experiment 1 regardless of whether static deciders are included or excluded. Please see below:

Original analyses (pg. 13):

“As predicted, selfish participants showed a bias towards self-serving memory errors... ($V = 335.5$, $p = .003$, $d = .34$, $\delta = .23$; Fig. 1A), whereas generous participants showed no such difference ($V = 353$, $p = .54$, $d = .13$, $\delta = .05$). However, there was no significant difference in the extent of this bias between selfish participants and generous participants ($W = 1292.5$, $p = .23$, $d = .19$, $\delta = .13$). Next, we tested if the two groups differed in memory accuracy – or the absolute size of their memory errors. We found no significant difference between selfish and generous behaving participants in memory accuracy ($W = 1730.5$, $p = .12$, $d = .21$, $\delta = .17$ ”

New analyses (in reply letter):

Experiment 1 results with static deciders excluded (from Table S6):

	test statistic	p	d	δ
Static deciders excluded (N=69)				
1. recalled vs. actual giving (all)	1268.5	.020	.28	.22
2. recalled vs. actual giving (selfish)	307.5	.004	.43	.34
3. recalled vs. actual giving (generous)	342	.45	.17	.11
4. memory errors (selfish vs. generous)	506.5	.31	.23	.14
5. memory accuracy (selfish vs. generous)	739.5	.076	.24	.25

”

However, upon further inspection of the table for Experiment 2 (Table S7), we observed that whereas our memory accuracy effect is significant in our main analysis, with no exclusions, and with non-givers excluded, the effect is non-significant when excluding static deciders. We thank for the reviewer for encouraging us to look for other

inconsistencies. We have updated the following text on pg. 4 of the supplemental material:

“Experiment 2

Main results (with no exclusions, non-givers excluded, and static deciders excluded). We also repeated our main analyses of Experiment 2 with static deciders excluded (N = 129), non-givers excluded (N = 184), as well as with no exclusions (N = 243). As with Experiment 1, the results were consistent with those reported in the main text.

We again found that participants showed a systematic bias towards self-serving memory errors, such that their recalled generosity was significantly greater than their actual generosity (Table S7.1). In addition, we replicated the finding that specifically violators recalled being significantly more generous than they actually were (Table S7.2), but not upholders (Table S7.3). Moreover, when comparing violators and upholders directly, the former showed a significantly greater bias toward self-serving memory errors (Table S7.4). We also found that violators were less accurate overall than upholders when making no exclusions and when excluding non-givers (Table S7.5). However, this latter effect was no longer significant when excluding static deciders (\$p = .11\$, \$d = .26\$, \$\delta = .16\$ ).”

3.3: You spotted a typo in Table S6.

Response:

Thank you for catching this, we have now fixed this error.

3.4: You remain concerned that differences in variance of the decisions made by violators vs. upholders (or exceders) could explain our key results.

“c. The authors took a clever, second approach to ruling out decision variance as an alternative explanation: comparing norm violators to norm exceders. Despite both deviating from their stated norms, these groups showed different levels of memory (with forgetting occurring more in the violators). This also helps somewhat, but unfortunately violators’ decisions still varied to a greater degree than exceders, meaning that this analysis can’t fully address the concern either.

Overall, I think the authors have done what they can on this point given the existing data, but there still seems to be room for the possibility that the sets of the decisions were just more difficult to remember or to average together. I continue to be hesitant to suggest an additional study that could rule this out more directly (e.g., as described in my review of the original manuscript), but at the very least I do suggest that this be noted as an alternative that can’t be fully ruled out under the circumstances, which the authors have added to the discussion”.

Response:

We agree that these analyses could not definitively rule out the reviewer’s concern. Inspired by the reviewer’s suggestion, we therefore conducted two additional experiments with 2,191 participants. Please see below for more details:

Experiment 3 showed that even when incentivized with a monetary bonus that directly scaled with memory accuracy, violators still systematically make self-serving memory errors. These results are consistent with the possibility of motivated misremembering. However, they only demonstrate this phenomenon indirectly. Indeed, a motivated account of misremembering would receive compelling evidence over a rational inference account with direct evidence for two key conditions.

First, motivated misremembering should be *motive-dependent*. That is, misremembering should occur only when people have a motive to misremember, and should cease to occur if motives to misremember are removed. In particular, a rich, long standing literature on motivated cognition suggests that when people no longer view themselves as *personally responsible* for a dissonance-inducing action, they no longer experience dissonance motivation (Cooper & Fazio, 1984; Gosling et al., 2006; Scher & Cooper, 1989; Wicklund & Brehm, 1976). Indeed, feelings of responsibility for an action serve as a bridge between the action and one's self-concept (Thibodeau & Aronson, 1992). As such, only if people feel personally responsible for their fairness violation should they experience a moral self-threat, and become motivated to reduce it via misremembering.

Second, motivated misremembering should be largely *choice-independent*. One possible alternative explanation for our findings is that there is something about the pattern of violators' realized choices that makes those patterns more prone to being misremembered—for instance, containing more variable transfer amounts. Thus, it is important to rule out the possibility that the sets of choices made by violators are just inherently more prone to being misremembered than the sets of choices made by upholders.

In a final pair of experiments (Experiments 4a and 4b), we examined these two hypotheses to directly test the motivated nature of misremembering. In Experiment 4a, an initial wave of participants (*free-choice* deciders) freely decided how to split money with anonymous partners, and later recalled their choices. In Experiment 4b, a second wave of participants (*forced-choice* deciders) were each yoked to a randomly selected free-choice decider from Experiment 4a, and were forced to make (and later recall) the exact same set of transfer choices as their yoked free-choice decider (See Figure S6). Crucially, in Experiment 4b, forced-choice deciders (and receivers) were instructed that the decider had *no responsibility* for their transfer choices—thereby removing any incentive to misremember their choices. Moreover, we confirmed that feelings of personal responsibility were lower among forced-choice (versus free-choice) deciders.

We made the following key predictions. Based on the motive-dependence hypothesis, we predicted that those who freely violated their own standards would make self-serving memory errors, whereas those who were forced to violate their own standards and perceived no personal responsibility for their actions would show no such bias toward self-serving memory errors.

Similarly, based on the choice-independence hypothesis, we predicted that participants asked to recall the choice sets freely made by violators would be no more likely to make self-serving memory errors than participants asked to recall the choice sets freely made by upholders.

Fig. 2. The top panel (A) shows an example of a yoked pair of deciders in Experiments 4a (free-choice) and Experiment 4b (forced-choice). The amount and order of all five transfers were the same. The bottom panel shows self-reported personal responsibility among deciders in Experiments 4a (free-choice; B) and Experiment 4b (forced-choice; C).

Fig. 4. Choice-independence of self-serving memory errors. **(A)** In Experiment 4a, free-choice violators tended to exhibit self-serving memory errors. **(B)** In contrast, in Experiment 4b, participants forced to make identical choices as violators exhibited no such bias toward self-serving memory errors. Error bars represent standard error of the mean (SEM). This figure is re-visualized with all raw data points in the *SI Appendix* (Fig. S4).

In summary, in our new experiments we found no evidence that misremembering was due to memory difficulty. We believe this new data allows us to firmly rule out the possibility that violators misremembered their choices more than upholders because of some inherent feature of their choice sets made them easier to misremember.

References

- Cooper, J., & Fazio, R. H. (1984). A new look at dissonance theory. In *Advances in experimental social psychology* (Vol. 17, pp. 229–266). Elsevier.
- Dana, J., Weber, R. A., & Kuang, J. X. (2007). Exploiting moral wiggle room: Experiments demonstrating an illusory preference for fairness. *Economic Theory*, *33*(1), 67–80.
- Exley, C. L. (2015). Excusing selfishness in charitable giving: The role of risk. *The Review of Economic Studies*, *83*(2), 587–628.
- Feiler, L. (2014). Testing models of information avoidance with binary choice dictator games. *Journal of Economic Psychology*, *45*, 253–267.
- Gosling, P., Denizeau, M., & Oberlé, D. (2006). Denial of responsibility: A new mode of dissonance reduction. *Journal of Personality and Social Psychology*, *90*(5), 722.
- Kunda, Z. (1990). The case for motivated reasoning. *Psychological Bulletin*, *108*(3), 480.
- Larson, T., & Capra, C. M. (2009). Exploiting moral wiggle room: Illusory preference for fairness? A comment. *Judgment and Decision Making*, *4*(6), 467.
- MacLeod, M. D., & Macrae, C. N. (2001). Gone but Not Forgotten: The Transient Nature of Retrieval-Induced Forgetting. *Psychological Science*, *12*(2), 148–152.
<https://doi.org/10.1111/1467-9280.00325>
- Macrae, C. N., & MacLeod, M. D. (1999). On recollections lost: When practice makes imperfect. *Journal of Personality and Social Psychology*, *77*(3), 463–473.
<https://doi.org/10.1037/0022-3514.77.3.463>
- Malle, B. F. (2006). The actor-observer asymmetry in attribution: A (surprising) meta-analysis. *Psychological Bulletin*, *132*(6), 895.
- Mazar, N., Amir, O., & Ariely, D. (2008). The dishonesty of honest people: A theory of self-concept maintenance. *Journal of Marketing Research*, *45*(6), 633–644.
- Ross, M., & Wilson, A. E. (2003). Autobiographical memory and conceptions of self: Getting better all the time. *Current Directions in Psychological Science*, *12*(2), 66–69.

- Scher, S. J., & Cooper, J. (1989). Motivational basis of dissonance: The singular role of behavioral consequences. *Journal of Personality and Social Psychology*, 56(6), 899.
- Shalvi, S., Gino, F., Barkan, R., & Ayal, S. (2015). Self-serving justifications: Doing wrong and feeling moral. *Current Directions in Psychological Science*, 24(2), 125–130.
- Stanley, M. L., Henne, P., Iyengar, V., Sinnott-Armstrong, W., & De Brigard, F. (2017). I'm not the person I used to be: The self and autobiographical memories of immoral actions. *Journal of Experimental Psychology: General*, 146(6), 884.
- Thibodeau, R., & Aronson, E. (1992). Taking a closer look: Reasserting the role of the self-concept in dissonance theory. *Personality and Social Psychology Bulletin*, 18(5), 591–602.
- Tulving, E. (1972). Episodic and semantic memory. *Organization of Memory*, 1, 381–403.
- Van den Berg, R., Shin, H., Chou, W.-C., George, R., & Ma, W. J. (2012). Variability in encoding precision accounts for visual short-term memory limitations. *Proceedings of the National Academy of Sciences*, 109(22), 8780–8785.
- Walster, E., Berscheid, E., & Walster, G. W. (1973). New directions in equity research. *Journal of Personality and Social Psychology*, 25(2), 151.
- Wicklund, R. A., & Brehm, J. W. (1976). *Perspectives on cognitive dissonance*.

Reviewers' comments:

Reviewer #2 (Remarks to the Author):

This is now the third time I've been asked to review this manuscript. My main concern with the last version of this experiment was the lack of any experiment that provided causal evidence for a motivated reasoning mechanism. The experiments added to this manuscript now do that in a reasonable way. My other concerns were relatively more minor and should not hold up publication of this manuscript. I think the paper has been made notably stronger by improving its scholarship and by providing better causal evidence of the underlying mechanism. I appreciate the authors' responsiveness to the concerns raised and congratulate them on a solid piece of research.

Reviewer #3 (Remarks to the Author):

My sense based on the last version of the manuscript was that the authors had done everything they could given their data, but that, at the same time, those data still left open at least one alternative interpretation of their findings: reduced memory for the choices of norm violators could have been due the lower memorability of the sets of decisions by those participants (relative to non-violator participants).

In this revision, the authors include new data from two studies in an effort to address this alternative interpretation directly. In one (4a), participants decided freely; in another (4b), participants' choices were forced, thereby creating patterns of responses that mirrored those of the free deciders in the first experiment.

When the authors described the overall results of these new experiments, they seemed incredibly promising. Since reading through the details, two issues have nagged. My hope/expectation is that the authors could probably address these rhetorically. The data are what they are, and it would just be useful to describe them in as true a fashion as possible.

One issue is that, if I understand correctly, there are indeed more memory errors by the forced deciders who were paired with the violators' (vs upholders') choices, consistent with the idea that these sets of decisions were intrinsically more difficult to remember. The memory errors weren't more likely to be self-serving, which the authors take to be important. But the fact that memory errors in forced-choice participants weren't biased doesn't mean that difficulty remembering couldn't have played a key role in the memory bias of the freely violating participants. Specifically, it seems difficult to disentangle a strong motivated misremembering account of the norm violators' memory from an account on which they simply don't remember well, due to the complexity of the data, and then fill in the missing information in a self-serving way. These two interpretations clearly both have roles for motivation and memory, but they put them at different places in the causal stream of events.

The other issue is that in the new pair of experiments, the authors do not report the results as a function of the manipulated variable – free vs forced choice. Rather, because the manipulation partly failed and some participants felt responsible even in the forced choice condition, the authors separately analyzed participants who “felt responsible” and those who did not. This is all described in a way that is consistent with the authors’ theoretical predictions. But by turning an experimental design into a correlational one, this raises the possibility that other factors might have differed between the participants who felt responsible and those who didn’t, diminishing somewhat the usefulness of this clever (albeit not totally successful) manipulation. Currently, the manipulation is presented as successful (p 32) but then the authors backtrack to say it wasn’t totally successful so they need to analyze the data in terms of participant subsets. If no overall difference in the constructs of interest was observed as a function of free- vs. forced-choice, it would be helpful if that were reported before proceeding with the subset analysis.

Reviewer Replies

Reviewer #2

R2.1: Reviewer 2 indicated that they were fully satisfied with our revised manuscript, in which we provided causal evidence for our claims.

Specifically, they wrote *“My main concern with the last version of this experiment was the lack of any experiment that provided causal evidence for a motivated reasoning mechanism. The experiments added to this manuscript now do that in a reasonable way. My other concerns were relatively more minor and should not hold up publication of this manuscript. I think the paper has been made notably stronger by improving its scholarship and by providing better causal evidence of the underlying mechanism. I appreciate the authors' responsiveness to the concerns raised and congratulate them on a solid piece of research.”*

Response:

We thank Reviewer 2 for their supportive words, and for the many theoretical challenges they raised during the review process. No doubt, our paper has greatly benefitted from having to carefully think about and address each of them.

Reviewer #3

R3.1: Reviewer 3 questioned whether we have ruled out the possibility that misremembering among violators was due to something intrinsic to the sets of decisions made by those participants. Specifically, they noted that there were “*more memory errors by the forced deciders who were paired with violators’ (vs upholders’) choices, consistent with the idea that these sets of decisions were intrinsically more difficult to remember*”. They also suggested that this might imply that memory difficulty aided free-choice violators in misremembering: “*it seems difficult to disentangle a strong motivated misremembering account of the norm violators’ memory from an account on which they simply don’t remember well, due to the complexity of the data, and then fill in the missing information in a self-serving way.*”

Response:

We thank Reviewer 3 for encouraging us to further explore this possibility. There are two distinct points here, and we will address each in turn. First, as noted in Reviewer 3’s original review, and revisited here, is the question of whether something intrinsic to the choice sets of violators drives their misremembering. Second, is the question of whether greater memory inaccuracy among force-choice violators is related to motivated misremembering in free-choice violators.

1. Is misremembering among violators due to intrinsic features of their choices?

To address this point, we will first recap the two memory measures in our paper:

1. *self-serving memory errors*—the signed difference between recalled vs. actual choices
2. *memory inaccuracy*—the absolute difference between recalled vs. actual choices.
 - **note:** we now refer to this measure as *memory inaccuracy* (as opposed to memory accuracy) because higher values on the measure correspond to greater *inaccuracy*.

We report both of these measures in our paper for completeness. However, the goal of our paper is *not* to show that people who violate their norms simply show greater memory inaccuracy. Rather, our novel contribution is demonstrating that violators *misremember* being more generous than they actually were. To establish this, we specifically needed to show, as was done, that norm violators make more self-serving memory errors.

We highlight this because we believe Reviewer 3’s concern in their original review was that *self-serving memory errors* in violators could be due to features of their choice sets. Reviewer 3 suggested two ways to address this concern:

(a) examine whether the structure of violators' choice sets can explain the misremembering effect within our existing datasets, and (b) if the first option cannot rule this possibility out: run an additional 'observer' study to test whether observers also exhibit *self-serving errors* when given violators' choice sets. We followed suggestion (a) in our first revision, taking several measures to rule out the possibility that intrinsic differences in violators' choice sets explain why they make more self-serving memory errors than upholders:

1. We controlled for decision variance, among other factors, in all key regression models, and found the fairness violations still were a key predictor of misremembering.
2. We re-ran our analyses with *static deciders* excluded (i.e., removing participants who made the same choice every time), and again showed that violators made significantly more self-serving memory errors than upholders.
3. We re-ran our analyses with *norm exceeders* (i.e., excluding those upholders whose average generosity matched their norm). Crucially, exceeders showed similar levels of decision variance as violators. When comparing exceeders with violators, we showed that only the latter made self-serving memory errors, while the former did not.

In their review of our first revision, Reviewer 3 accidentally misread the static decider analyses (#2 above) as yielding nonsignificant results. However, as we clarified in our recent appeal, the difference in misremembering between violators vs. upholders holds when excluding static deciders. Furthermore, Reviewer 3 suggested the exceeder analysis cannot fully address their concern as decision variance still was different between groups. However, in the static decider analyses, there is *no difference* between groups on decision variance in studies 3, 4a, and 4b (now shown in Table S14; p. 31).

Overall, Reviewer 3 seemed satisfied, writing that they are "*hesitant to suggest an additional study that could rule this out more directly...but at the very least I do suggest that this be noted as an alternative that can't be fully ruled out under the circumstances, which the authors **have added** to the discussion.*" (emphasis added)

Since our appeal involved conducting additional studies to address Reviewer 2's concerns, we designed them with Reviewer 3's suggestion (b) in mind—integrating their creative idea to: "*show a new group of "observer" participants the five decisions of various participants from the current studies...and then later ask these observers the average percentage given. If observers also show more **memory distortions** for the sets of decisions made by the ultimate violators than by the ultimate upholders (**by overestimating violators' generosity**), this would point to an alternative explanation, potentially having to do with structure of those sets. On the other hand, to the extent that observers did not show the **memory bias** shown by violators, this would help to rule out the possibility that there's something about the structure of those sets that affects how participants*

*integrate the choices, thereby helping to rule in the authors' favored explanation that there's something about the psychology of the norm violation that **distorts people's memory**.*" (emphasis added)

We have bolded portions of Reviewer 3's original review, as they indicate a clear focus on *motivated misremembering*, which, as we emphasized above, requires that we analyze self-serving memory errors, rather than memory inaccuracy. Our new experiment showed exactly what Reviewer 3 originally requested: that observers, unlike violators, exhibit no "memory bias" or "overestimating".

Together, we believe these findings directly address this original concern that misremembering among violators could reflect an artifact of the structure of violators' choice sets.

Our interpretation of Reviewer 3's recent review is that they have shifted their focus, such that memory inaccuracy (rather than self-serving memory errors) is now their key concern. We turn to this issue next.

2. Is greater memory inaccuracy among force-choice violators related to motivated misremembering in free-choice violators?

In their most recent review, Reviewer 3 suggests that finding greater memory inaccuracy among *forced-choice* violators poses a new issue.

To confirm, this finding was expected. It is consistent with the results of our earlier experiments, and follows from basic principles of memory. Across all studies, *decision variance* was related to greater memory inaccuracy (correlations ranged from $r = .57-.71$), and violators reliably show greater decision variance than upholders (Table S3). This is because upholders often matched their norm, making the exact same decision 5 times. Thus, upholder choice sets, more often than violator choice sets, were trivially easy to recall. And it follows that memory inaccuracy should be lower for those recalling violator choice sets, regardless of whether one freely made the choices or not.

Reviewer 3 suggested that greater memory inaccuracy among forced-choice violators is consistent with the idea that self-serving memory errors among *free-choice* violators is *choice-dependent*—i.e. they misremember due to their choices being harder to recall.

However, this proposal is at odds with our evidence that misremembering persists when controlling for a key source of memory difficulty. For instance, when we exclude static deciders (thereby reducing or wiping out the decision variance gap), memory inaccuracy differences between violators and upholders shrink or disappear in all studies, whereas self-serving memory errors remain the same (See Tables S9-S13).

That said, Reviewer 3's proposal can be tested more directly. If greater memory inaccuracy among forced-choice violators arises from the same intrinsic choice features that also allowed free-choice violators to make self-serving memory errors, then we should see that greater memory inaccuracy among forced-choice violators should predict whether free choice violators make self-serving memory errors or not. Below we provide a simple test of this question, which we have added to the Supplemental Results (p. 8):

“A further test of the choice independence of misremembering: does memory inaccuracy among forced-choice violators (Experiment 4b) predict self-serving memory errors in free-choice violators (Experiment 4a)? Since forced-choice violators showed greater memory inaccuracy than upholders—just as free choice violators did—it is possible that the memory errors of forced-choice violators might reflect the ‘true’ memorability of free violators choices, and that this true level of memorability could be related to free choice violators tendency to make self-serving memory errors. A reviewer suggested that it is important to rule out this possibility, as the same cause of memory inaccuracy in forced-choice violators could be related to self-serving memory errors in yoked, free-choice violators.

That is, among those free-choice deciders who made memory errors, can the direction of their errors (self-serving versus self-defeating) be predicted by memory inaccuracy in yoked forced-choice deciders? More importantly, does it have predictive value above and beyond whether or not a free-choice decider violated their norm?

We tested this possibility with a logistic regression model in which memory error direction (self-serving [1] vs. self-defeating [0]) was predicted by (i) norm violation status (violated [1] vs. upheld [0]), (ii) memory inaccuracy of yoked forced-choice deciders, and the interaction of (i) and (ii). The results supported the choice-independence account: while the model explained a significant degree of the variance ($R^2 = .036$, $F(3,241) = 3.00$, $p = .031$), the only significant predictor was the free-choice decider's norm violation status ($\beta = .083$, $p < .004$). Indeed, neither the memory errors of yoked forced-choice deciders ($\beta = -.020$, $p = .57$), nor the interaction of norm violation status and these yoked memory errors ($\beta = .011$, $p = .73$), significantly predicted the occurrence of self-serving memory errors in free-choice deciders.”

These data suggest that greater memory inaccuracy among forced-choice violators does not predict misremembering in free-choice violators—consistent with the view that motivated misremembering is choice-independent. To make this clearer for readers, we've elaborated on this point in the Supplemental Discussion (p. 9):

“Note on choice variability and memorability. Above we show that our findings remain consistent when only including those who made dynamic decisions (i.e., when we exclude static deciders). This supports the idea that motivated misremembering cannot be simply explained by choice set memorability. This possibility seemed plausible given that violators showed greater memory inaccuracy. Yet whereas excluding static deciders—and thus balancing the decision variance gap between groups—greatly diminishes memory inaccuracy differences between violators and upholders, the motivated misremembering difference remains the same. In addition, motivated misremembering among violators disappears when personal responsibility is removed, which has a negligible influence on memory inaccuracy. This pattern of results is consistent with the view that group differences in decision variance drive memory inaccuracy differences, but they do not drive motivated misremembering. Further support for the idea that motivated misremembering is choice independent was garnered by demonstrating that memory

errors among forced-choice deciders do not predict self-serving memory errors in free-choice deciders.”

We have also referenced this finding in the main text, within the results of Experiment 4 (p. 37):

“While those paired with violators tended to have less accurate memories in general ($W = 23396.5$, $p < .001$, $d = 0.43$, $\delta = 0.35$), memory inaccuracy in forced-choice violators did not predict self-serving memories in yoked free-choice violators (see *SI Appendix*).”

R3.2: You asked us to report our main comparison between forced upholders and violators before conducting subset analyses, and to note the limitations of this approach.

Response:

Thank you for suggesting this. In our prior submission, the results for the forced-choice study were presented in a way that we thought would be easiest for readers to interpret. We also note that we specifically included the self-reported responsibility measure in our experiments because we anticipated before running our studies that some forced-choice violators may nevertheless feel responsible for violating their norm and be motivated to misremember. This prediction was based on classic studies of cognitive dissonance and obedience to authority showing that while following orders or instructions can reduce feelings of responsibility, it does not eliminate them entirely (e.g., Gosling et al., 2006; Mantell & Panzarella, 1976).

That said, we are fully on board with prioritizing transparency. As such, we have revised our initial reporting of the free/forced choice manipulation results as follows (p. 32):

“To evaluate the efficacy of our manipulation, we first assessed how *responsible* participants perceived themselves to be for their actions under free-choice (Experiment 4a) and forced-choice (Experiment 4b) conditions. As predicted, free-choice participants reported a high degree of responsibility for their actions ($M = 6.28$, $SD = 1.18$), with 64% ($N = 452/709$) reporting the highest level of responsibility possible for their actions (7 = “Extremely responsible”; *SI Appendix*, Fig. S6A). In contrast, forced-choice participants reported a low degree of responsibility for their actions ($M = 2.14$, $SD = 2.03$), with 71% ($N = 409/580$) reporting the lowest level of responsibility possible for their actions (1 = “Not at all responsible”; *SI Appendix*, Fig. S6B). The difference in ratings between these two groups was significant ($W = 24374223.5$, $p < .001$, $d = 2.57$, $\delta = 83$). Nevertheless, some participants in the forced-choice condition indicated they felt some degree of responsibility for their choices despite not making them freely. This is plausible in the context of our studies because participants ultimately had to still move the slider to register the predetermined choice.”

In addition, we have now revised the results section to report the findings of the forced choice condition in full, before moving on to subset analyses (p. 33-34):

“In Experiment 4b, forced-choice participants did not show a significant bias towards self-serving memory errors ($V = 16087.5$, $p = .16$, $d = .11$, $\delta = .05$; *SI Appendix*, Table S1). Further analyses revealed that similar to free-choice upholders, forced-choice upholders ($N = 362$, average generosity = 46.7%) did not exhibit self-serving memory errors ($V = 3393$, $p = .14$, $d = .02$, $\delta = -.02$). Forced-choice violators ($N = 217$, average generosity = 20.95%)—despite not freely choosing to violate their norm—nevertheless did recall on average being more generous than they actually were ($V = 4580.5$, $p < .001$, $d = .28$, $\delta = .15$). However, the magnitude of misremembering was smaller in forced-choice violators ($d = .28$), than free-choice violators ($d = .40$). We also found that forced-choice violators showed a greater bias toward self-serving memory errors than upholders ($W = 33105.5$, $p < .001$, $d = .34$, $\delta = .16$), and showed greater inaccuracy than upholder ($W = 31604.5$, $p < .001$, $d = .23$, $\delta = .20$), though again these effects were smaller than those in the free choice condition ($d = 0.43$ and $d = 0.41$, respectively).

Thus, while our forced choice manipulation eliminated the main effect of misremembering, there remained a significant, albeit diminished, misremembering effect for forced-choice violators. One possibility is that, despite our instructions that sought to minimize feelings of responsibility in the forced choice experiment, some forced-choice violators may have still felt some degree of personal responsibility, and thus were motivated to misremember. This prediction follows from work showing that while following orders or instructions can reduce feelings of responsibility, it does not eliminate them entirely (e.g., Gosling et al., 2006; Mantell & Panzarella, 1976). Supporting our prediction, around 30% of forced-choice participants reported feeling some degree of responsibility for their choices.

Crucially, the motive dependence hypothesis predicts that the forced choice manipulation should eliminate motivated misremembering *only* in those violators who viewed themselves as *not personally responsible* for their actions, whereas forced-choice violators who viewed themselves as personally responsible should remain motivated to make self-serving memory errors. To test this possibility, we independently assessed the memories of those who reported being “not at all responsible” for their actions (i.e., 1 out of 7 on our personal responsibility measure; $N = 408$), and those who self-reported some degree of personal responsibility for their actions (i.e., greater than 1 out of 7 on our personal responsibility measure; $M = 4.87$, $SD = 1.84$; $N = 171$).”

In addition, following your suggestion, we added the following note to the general discussion noting the limitations of using our subset approach (p. 38):

“Specifically, these findings suggest that those who violate (as opposed to uphold) their personal standards misremember the extent of their selfishness. Moreover, they highlight the key motivational role of perceived responsibility for norm violations—consistent with classic accounts from social psychology (Wicklund & Brehm, 1976). However, since we focused specifically on those who reported no responsibility, it is also possible that other factors might have differed between the participants who felt responsible and those who did not.”

Again, we thank the reviewer for helping us improve the clarity of our manuscript. We hope that the edits and clarifications outlined above will satisfy the reviewer and editor.

Reviewer #3 (Remarks to the Author):

I have no further comments on this manuscript.